

# Holocene sea level and environmental change at the southern Cape - an 8.5 kyr multi-proxy paleoclimate record from lake Voëlvlei, South Africa

Paul Strobel[1], Marcel Bliedtner[1], Andrew S. Carr[2], Peter Frenzel[3], Björn Klaes[4], Gary Salazar[5], Julian Struck[1], Sönke Szidat[5], Roland Zech[1], Torsten Haberzettl[6]

[1]Physical Geography, Institute of Geography, Friedrich Schiller University Jena, Jena, Germany
[2]School of Geography, Geology and the Environment, University of Leicester, Leicester, UK
[3]Institute of Geosciences, Friedrich Schiller University Jena, Jena, Germany
[4]Department of Geology, University of Trier, Trier, Germany
[5]Department of Chemistry and Biochemistry and Oeschger Centre for Climate Change Research, University of Bern, Bern, Switzerland
[6]Physical Geography, Institute of Geography and Geology, University of Greifswald, Greifswald, Germany

*Correspondence to*: Paul Strobel (paul.strobel@uni-jena.de)

**Abstract.**

South Africa is a key region for paleoclimate studies reconstructing and understanding past changes in atmospheric circulation, i.e., temperate Westerlies and tropical Easterlies. However, due to the scarcity of natural archives, the environmental evolution during the late Quaternary remains highly debated. Many archives that are available are peri-coastal lakes and wetlands and sea level changes during the Holocene often overprinted the paleoenvironmental signals in these archives. This study presents a new record from the coastal wetland Voëlvlei, which is an intermittent lake situated in the year-round rainfall zone (YRZ) of South Africa at the southern Cape coast. It presents an ideal archive to investigate both sea level and environmental changes. A 13 m-long sediment core was retrieved from Voëlvlei and analysed using a multi-proxy approach. The chronology reveals a basal age of 8,440 $^{+200}/_{-250}$ cal BP. Paleoecological and elemental analyses indicate marine intrusions from 8,440 to 7,000 cal BP with a salinity optimum at 7,030 $^{+150}/_{-190}$ cal BP. Since 6,000 cal BP, silting up has been causing an intermittent freshwater lake.

Inferred from changes in allochthonous input, $\delta^{13}C_{n\text{-alkane}}$ and $\delta^{2}H_{n\text{-alkane}}$ increasing moisture is observed from 8,440 $^{+200}/_{-250}$ cal BP. The $\delta^{2}H_{n\text{-alkane}}$ record provides new evidence in contribution of different precipitation sources throughout the record with contributions from both Westerlies and Easterlies from 8,440 to 7,070 cal BP. Westerlies dominate from 7,070 to 6,420 cal BP followed by a distinct shift to an Easterly-dominance at 6,420 cal BP. An overall trend to a Westerly- lasting until 2,060 cal BP is followed by a trend towards an Easterlies-dominance, but both phases show several climatic spikes. Those spikes are also evident in other regional studies highlighting that the source and seasonality of precipitation has a mayor role for the hydrological balance. By comparing the Voëlvlei record with other regional studies, a similar trend in the overall moisture evolution along the southern Cape coast is inferred during the past 8.500 yrs.



## 1 Introduction

A record-breaking drought occurred in South Africa from 2015 to 2017 and future climate projections are even worse for large
parts of the country (Engelbrecht and Engelbrecht, 2016; Engelbrecht et al., 2011). To make climate models and predictions
of further hydrological change more reliable, robust paleoclimate reconstructions using direct hydrological proxies are
necessary, which are hitherto very rare in South Africa (Haberzettl et al., 2014). Southern Africa's past and present climate
has been driven by complex interactions between two major oceanic and atmospheric circulation systems, i.e., the Benguela
and Agulhas current, and the Westerlies and Easterlies (Tyson and Preston-Whyte, 2000) (Fig. 1 A). Today, three major
rainfall zones occur in South Africa. While the eastern and central parts of the country receive most rainfall ($> 66\%$) from
tropical moisture-bearing atmospheric circulation systems during austral summer (Summer Rainfall Zone, SRZ), a narrow belt
along the west coast receives most rainfall ($> 66\%$) from temperate Westerlies during the austral winter (Winter Rainfall Zone,
WRZ) (Fig. 1 A). An intermediary area between the SRZ and WRZ receives rainfall from both systems throughout the year
(Year-round Rainfall Zone, YRZ) (Fig. 1 A) (Engelbrecht et al., 2015; Scott and Lee-Thorp, 2004), and this includes the
southern Cape coast, which is the focus of this study.

The YRZ has been the focus of most paleoenvironmental and associated -paleoclimatic research. There, the southern Cape
coast and especially the Wilderness area with its numerous coastal lakes including such as Bo Langvlei (du Plessis et al., 2020),
Eilandvlei (Kirsten et al., 2018a; Kirsten et al., 2018b; Quick et al., 2018; Reinwarth et al., 2013; Wündsch et al., 2018;
Wündsch et al., 2016b), Groenvlei (Martin, 1959, 1968; Wündsch et al., 2016a) and Swartvlei (Birch et al., 1978; Haberzettl
et al., 2019) has yielded multiple paleoenvironmental records (Fig. 1 B). These coastal lakes have formed between large coastal
dune cordons that lie parallel to the coast. However, the terrestrial climate signals in these coastal archives are often overprinted
by marine water intrusions induced by relative sea level change during the Holocene (Martin, 1959, 1968; Reinwarth et al.,
2013; Wündsch et al., 2018; Wündsch et al., 2016a) or anthropogenic impact (Haberzettl et al., 2019). Further
paleoenvironmental information from this area is available from e.g., peatlands (Quick et al., 2016; Strobel et al., 2019) and
speleothems (Braun et al., 2018; Braun et al., 2020; Talma and Vogel, 1992), rock hyrax midden (Chase et al., 2019; Chase et
al., 2020; Chase et al., 2017; Chase et al., 2018; Chase et al., 2015) and marine sediments (Hahn et al., 2017) (Fig. 1 A).
However, the climate evolution of South Africa is still debated, reflecting potential spatial variability in climate drivers at the
regional scale, and due to the application of different methodological approaches in various studies (Chase and Quick, 2018;
Strobel et al., 2019). Consequently, the understanding of environmental dynamics and changing interactions between tropical
and temperate climate systems affecting the YRZ is limited.

Compound-specific stable isotope analyses of hydrogen and carbon isotopes of long-chain $n$-alkanes ($\geq C_{25}$; $\delta^2 H_{n\text{-alkane}}$,
$\delta^{13}C_{n\text{-alkane}}$) are valuable proxies that complement established methodological approaches related to paleoenvironmental and
paleohydrological changes in sediment archives, e.g., grain sizes, geochemistry and pollen. Long-chain $n$-alkanes are leaf
waxes produced by higher terrestrial plants and serve as valuable biomarkers, as they remain well preserved in soils and
sediments over millennia because of their low water solubility and high resistance against degradation (Eglinton and Eglinton,





2008; Sachse et al., 2012; Sessions, 2016). In South Africa, the $\delta^2 H_{n\text{-alkane}}$ signal shows the potential to reconstruct the isotopic signal of precipitation and thus directly refers to the precipitation source (Herrmann et al., 2017; Strobel et al., 2020). Although, $\delta^2 H_{n\text{-alkane}}$ has hitherto rarely been used in terrestrial archives at the southern Cape coast (Strobel et al., 2019).

$\delta^{13}C_{n\text{-alkane}}$ is a suitable proxy to infer past changes in the vegetation composition (e.g., Diefendorf and Freimuth, 2017) as well

as variations in plant water use efficiency and thus drought stress (Diefendorf and Freimuth, 2017; Struck et al., 2020). Therefore, using the climatic information from both leaf wax isotopes enables climate reconstructions based on the isotopic signal of precipitation, which can reflect local water availability. Further, leaf wax-derived $n$-alkanes can be used as a chronological marker since they can be dated using $^{14}C$ analyses (Bliedtner et al., 2020; Bliedtner et al., 2018; Douglas et al., 2014; Gierga et al., 2016; Haas et al., 2017).


[Figure 1]

Here we present a sediment record from Voëlvlei, today an intermittent lake, to reconstruct past sea level and environmental changes at the southern Cape coast of South Africa. Therefore, a multi-proxy approach has been applied to the sediments

comprising compound-specific stable isotope analyses on leaf waxes as well as established sedimentological, and (in)organic elemental, and paleoecological analyses on fossil associations. Specifically we aim at:

    i)   establishing a robust chronology based on diverse dating approaches on different sediment compounds,

    ii)  disentangling marine and climate influences during the development of lake Voëlvlei, and

    iii) inferring variations in local moisture availability and the source of precipitation.

**2 Site description**

Voëlvlei is situated ca. 40 km east of Still Bay and 30 km west of Mossel Bay, at an elevation of 5 m above present sea level (a.s.l.) ~10 km inland of the Indian Ocean coast. Today, Voëlvlei has an area of 3.8 km² (max. length 4.2 km; max. width 0.7 km) (Database: SRTM 1 arc-second) and the catchment has an area of 165 km² (Database: SRTM 1 arc-second) (Fig. 1 C). The catchment comprises altitudes between 5 and 333 m a.s.l. (Database: SRTM 1 arc-second) and is drained by one main

ephemeral river, which enters Voëlvlei in the north. A barrier elevated up to 17 m a.s.l. (Database: SRTM 1 arc-second) defines the southern border of the Voëlvlei catchment. Voëlvlei has one intermittent outflow in the southwest at 6 m a.s.l. (SRTM 1 arc-second) and drains into the Gouritz river (Fig. 1 C).

      The geology is characterised by Palaeozoic quartzites of the Table Mountain Group (Cape Supergroup), mudrock-sandstones of the Bokkeveld Group (Cape Supergroup), Mesozoic mudrock-sandstone conglomerates (Uitenhage Group), and

Cenozoic limestone-sandstone conglomerates (Bredasdorp Group) (Johnson et al., 2006). Soils have high aluminium and iron concentrations and are mostly Cambisols and Leptosols (Fey, 2010; Zech et al., 2014).



The potential natural vegetation consists of variations of Fynbos and only small areas along the main drainage system would be covered by Albany Thicket (Mucina and Rutherford, 2006). Today, large areas of the catchment are used for agriculture and some pastures persist. The steep slopes of the drainage system are mainly unmanaged and covered by plant communities of the Fynbos and Albany Thicket.

Mean annual precipitation at the study site is 450 mm·a$^{-1}$ (Fick and Hijmans, 2017) and rainfall is almost equally distributed throughout the year. Winter precipitation is linked to the temperate Westerlies related to the Atlantic ocean as moisture source, and summer precipitation is associated with the tropical Easterlies and the Indian Ocean as moisture source (Engelbrecht and Landman, 2016). Moreover, orographic rainfall occurs from local sources due to onshore flows related to ridging anticyclones (Weldon and Reason, 2014). The isotopic composition of precipitation ($\delta^2H_p$) is $^2H$ depleted during winter and $^2H$ enriched during summer periods, with a modelled annual mean of -13 ± 1 ‰ (Table 1) (Bowen, 2018; Bowen et al., 2005; Braun et al., 2017; Harris et al., 2010). Mean annual temperature is 17.6 °C and slightly higher temperatures during summer (22° C) lead to semi-arid climatic conditions at the study site today (Fick and Hijmans, 2017).

*[Table 1]*

## 3 Material and Methods

For this study, the 13 m long sediment core VOV16 was retrieved from Voëlvlei (34.259°S; 21.826°E) (Fig. 1 C) in 2016 using a motor hammer coring system (inner core diameter 5 cm) and transported to the Physical Geography laboratory of the Friedrich Schiller University Jena where it was stored dark and cool at +4°C until processing. Cores were split, photo-documented and described in detail concerning sedimentological properties and sediment colour in the laboratory.

### 3.1 Chronology

### 3.1.1 Radiocarbon dating of macro particle, bulk TOC and n-alkane samples

The chronology of the sediment record is based on $^{14}C$-ages from one organic macro particle, three charcoal samples, 15 bulk organic samples and seven *n*-alkane samples (compound-class). 12 of the bulk organic samples and the organic macro particle were analysed with the AMS at the Poznan Radiocarbon Laboratory, Poland. Three bulk organic, three charcoal and seven *n*-alkane samples (cf., section 3.7 for sample extraction prior to measurement) were analysed with the Mini Carbon Dating System (MICADAS) AMS coupled to an element analyser (Ruff et al., 2010; Salazar et al., 2015; Szidat et al., 2014) at the LARA AMS Laboratory, University of Bern, Switzerland. $^{14}C$ results from the LARA AMS were reported as $F^{14}C$ and corrected for cross and constant contamination after Salazar et al. (2015).



### 3.1.2 OSL dating

A split (half) section of the core was sub-sampled under red light conditions in the University of Leicester luminescence dating laboratory. The upper 5-6 mm of the sediment surface was removed, and the core section was sampled over a depth range of 70 mm. Sediment within 6-7 mm of the core tube inner surface was not sampled but was used for an estimation of sample water content. This (dry) material, as well as material from the upper surface was homogenised and used for dose rate analysis. The sediment taken for equivalent dose analysis was soaked in sodium hexametaphosphate and then wet sieved. The core sediments yielded relatively limited amounts of sand sized material, most of which fell within the fine sand range ($< 100\,\mu m$). This necessitated the use of the fine sand range 55-90 µm for equivalent dose analysis. This material was then prepared using standard methods. This involved treatment with dilute (10%) hydrochloric acid and (32%) hydrogen peroxide. The sample was then dried, and density separated to isolate the $< 2.7$ g·cm⁻³ and $> 2.58$ cm⁻³ fraction, before etching for 45 minutes in 48% hydrofluoric acid, washing in hydrochloric acid and dry sieving.

Dose rates were determined using the remaining core material via inductively coupled plasma mass spectrometry (ICP-MS) for U and Th and ICP-OES for K analyses at the University of Leicester. The concentrations of U, Th and K were converted to annual dose rates following Guérin et al. (2011) with corrections for grain size (Mejdahl, 1979), water content (Aitken, 1985) and HF etching (Bell, 1979). Cosmic dose rates were determined using the reported sample depth following Prescott and Hutton (1994) with a 5% relative uncertainty included. Final age uncertainties incorporate 3% relative uncertainties for the dose rate conversion factors, grain size attenuation factor, water attenuation and HF etching, propagated via standard methods. HF etching is assumed to have entirely removed the α-irradiated outer portion of the quartz grains. It was assumed that the as-measured water content was appropriate, with a 3% (absolute content) uncertainty propagated to the final dose rate uncertainty.

All luminescence measurements were performed on a Risø DA20 TL/OSL reader. Stimulation (40 s at 125°C) was provided by blue LEDS (stimulation wavelength 470 nm) with OSL signals detected with an EMI 9235QA photomultiplier tube via a U-340 detection filter. Laboratory irradiations were delivered by a ⁹⁰Sr beta source with a dose rate (at the time of measurement) ~7.58 Gy·min⁻¹. Sample equivalent doses ($D_e$) were determined using the Single Aliquot Regeneration (SAR) protocol (Murray and Wintle, 2000, 2003; Wintle and Murray, 2006). All single aliquot measurements were carried out on small (1 to 2 mm) aliquots, which given the grain size fraction analysed means there are likely more than 1000 grains per aliquot (e.g., Duller, 2008).

A dose recovery preheat experiment was used to assess the suitability of the SAR protocol in general, and the most appropriate preheating conditions. The overall dose recovery ratio across all preheating temperatures 160-260 °C was $0.97 \pm 0.03$ (n=22; zero overdispersion), with a ratio of $1.00 \pm 0.02$ (n=3) for the chosen preheat temperature combination of 240°C for ten seconds (natural regeneration points) and a 220 °C cut heat for the test dose measurements. All SAR analyses comprised a 7-regeneration point sequence, which included a repeated (recycling) regeneration dose point, and IR depletion regeneration dose point to check for K feldspar contamination (Duller, 2003) and a zero-dose point. The 4 (unique) point dose response

curve was generated using the initial 0.64 seconds of stimulation, with a background signal from the last 8 seconds. Analyses were carried out in the "Analyst" software. Dose response curves were fitted with saturating exponential fits with $D_e$
uncertainties incorporating counting statistics, curve fitting uncertainties and a 1% systematic uncertainty (Duller, 2007) (all calculated within the Risø Analyst software). The uncertainty in final $D_e$ estimate also incorporates an additional beta source calibration uncertainty (3%).

### 3.1.3 Age modelling

For the final age-depth modelling, all aforementioned results were used. Bulk organic and *n*-alkane [14]C-ages from the terrestrial
part of the sediment core were calibrated with the SHcal20 data calibration curve (Hogg et al., 2020), whereas [14]C-ages from the marine part of the record were calibrated with the Marine20 calibration curve (Heaton et al., 2020) and additionally corrected for a marine reservoir effect using a $\Delta R$ of $134 \pm 38$ [14]C yrs as previously reported by Wündsch et al. (2016b) (Table 1). For the compound-class *n*-alkane samples, the SHCal20 calibration curve (Hogg et al., 2020) was applied due to a predominance of terrestrial synthesized long-chain *n*-alkanes (C$_{29}$, C$_{31}$, C$_{33}$) in the samples (cf., section 4.3; Table 1). The
terrestrial organic macro particle and charcoal found in the marine part of the record were calibrated with the SHCal20 curve (Hogg et al., 2020) (Table 1). All calibrations were done with the online version of the Calib 8.2 software (Stuiver et al., 2020). The final age-depth relation was calculated with the R software package Bacon 2.4.3 (Blaauw and Christen, 2011), using the same calibration data sets.

### 3.2 Grain size analyses

For grain size measurement 50 mg sample aliquots were treated with $H_2O_2$ (10% p.A.) to remove organic matter, HCl (10% p.A.) for the destruction of carbonates, and $Na_4P_2O_7/Na_2CO_3$ for dispersion. The grain-size distribution of each sample was determined with a Laser Particle Sizer (FRITSCH ANALYSETTE 22 Microtec, FRITSCH, Germany) and a wet dispersion unit at the Physical Geography laboratory of the University of Greifswald. Before the measurement, samples were treated with ultrasound for 60 seconds and subsequently measured in duplicate. The grain size distribution is calculated in 99 classes
between 0.08 – 2,000 µm. The mean and median grain size as well as fractions of clay, silt and sand of each sample were calculated.

### 3.3 Paleontological analyses

A selection of 23 sediment samples representing all lithological units and focusing on the assumed transition from an estuarine to a freshwater environment was processed and analysed micropaleontologically. About 3-7 ml sediment from 1-cm thick
sediment slices of the core were washed with tap water through stacked sieves of 63 µm and 200 µm mesh size. After drying the sieve residues on a heating plate at ~50°C, all microfossils were picked under a low-power stereomicroscope and transferred to microfossil slides for later identification and counting. Microfossils, fragments of macrofossils and charcoal were documented semi-quantitatively as rare (1-2 specimens), common (>2), abundant (>10) and very abundant (>100).





Identification relies on Benson and Maddocks (1964), Martens et al. (1996) Fürstenberg et al. (2017) for Ostracoda and on

Schmitt-Sinns (2008) and Fürstenberg et al. (2017) for Foraminifera. Additionally, macrofossils found during the lithological description, sampling of the core and processing the microfossils were identified, counted and used for paleoecological interpretation. Those macrofossils are all snails identified relying mainly on Branch et al. (2010) and the mollusc section of the World Register of Marine Species (www.marinespecies.org). Bivalves could not be identified because they occurred only as fragments. Paleoecological information was drawn from the papers listed above for identification plus Murray (2006) and

Kirsten et al. (2018a). All paleontological material is stored in the collection of the authors at Friedrich Schiller University Jena and will be transferred to the South African Museum of Natural History in Cape Town later.

### 3.4 Elemental analyses

XRF data were collected every 1 cm down-core using two generator settings (30 kV, 1 mA, 15 s; 15 kV, 0.2 mA, 15 s) for detection of different elemental groups with XRF Core Scanner II (AVAATECH Serial No. 2) at MARUM - University of

Bremen. The split core surface was covered with a 4 micron thin SPEXCerti Prep Ultralene1 foil to avoid contamination of the XRF measurement unit and desiccation of the sediment. The here reported data have been acquired by a Canberra X-PIPS Silicon Drift Detector (SDD; Model SXD 15C-150-500) with 150eV X-ray resolution, the Canberra Digital Spectrum Analyzer DAS 1000, and an Oxford Instruments 50W XTF5011 X-Ray tube with rhodium (Rh) target material. Raw data spectra were processed by the analysis of X-ray spectra by Iterative Least square software (WIN AXIL) package from Canberra Eurisys.

Presented data were normalised by elemental Zr counts and plotted as log-ratios, primarily to eliminate sediment matrix errors (water content, surface roughness and grain size variations) (Weltje and Tjallingii, 2008).

Moreover, 150 sample aliquots in an interval of 8 cm were freeze dried (-53 °C, for > 48 h), ground and sieved to a size < 40 µm. Aluminium (Al), calcium (Ca) and sodium (Na) concentrations were measured with an ICP-OES 725-ES (VARIAN, USA) at the Physical Geography laboratory of the Friedrich Schiller University Jena. 0.2 g of the samples was processed using

a microwave-assisted modified aqua regia digestion of 2 ml HCL (32% p.A.) and 4 ml $HNO_3$ (65% p.A.). Error estimates were based on triple measurements of three samples (VOV_243, VOV_651, VOV_1186; numbers indicate sample depth) (relative error: Al 3.7%; Ca: 5.4%; Na: 5.3%). Samples of the reference material LGC6 187 (river sediment) were measured as well to calculate the relative analytical error which was 1.1% for Al and Ca, and 10% for Na.

Total nitrogen (TN) and total carbon (TC) were analysed with a CNS analyser (EuroVector EA 3000, HEKAtech GmbH,

Germany) at the Physical Geography laboratory of the University of Greifswald. Concentrations of total organic carbon (TOC) were determined with the same device after treatment with 3% and 20% HCl at 80 °C to remove carbonates. Error estimates were based on triple measurements of 30 samples (mean relative error: N: 8.9%; TOC: 2.9%). Total inorganic carbon (TIC) was calculated as difference between TC and TOC.



### 3.5 Biogenic Silica (BiSi)

Biogenic silica (BiSi) concentrations were determined following Ohlendorf and Sturm (2008). BiSi and Al concentrations were measured using an ICP-OES 725-ES (VARIAN, USA) at the Physical Geography laboratory of the Friedrich Schiller University Jena. A correction factor of one (BiSi:Al; 1:1) was applied to account for dissolving aluminosilicates.

### 3.6 Powder X-ray diffraction (XRD)

The identification of the mineral composition of 50 powdered samples (e.g., Pecharsky and Zavalij, 2009) from representative
sediment core sections was carried out using an X-ray diffractometer (D8-Discover, Bruker AXS) equipped with a CuKα X-ray tube and a gas proportional counter (HI-STAR area detector, Bruker AXS) at Friedrich Schiller University Jena. The qualitative analyses and interpretation of the diffractograms was conducted at Trier University using Bruker DIFFRAC plus 5.0 software. The occurrence of specific mineral phases (low, medium, high) was roughly estimated based on XRD peak intensities in conjunction with the elaborated geochemistry of the investigated sediment sections. For all measurements the
quartz peak at 3.342 Å was accepted as internal standard.

### 3.7 Biomarker analyses of n-alkane distributions and stable isotope compositions

Total lipids of the sediment samples (14.5 – 31.4 g) were extracted with 40 ml dichloromethane (DCM) and methanol (MeOH) (9/1, v/v) using an ultrasonic bath over three 15 min cycles. The total lipid extract was separated by solid phase extraction using aminopropyl silica gel (Supelco, 45 µm) as stationary phase. The *n*-alkanes were eluted with 4 ml hexane and further
purified over coupled silvernitrate (AgNO$_3$-) – zeolite (Geokleen) pipette columns. The *n*-alkanes trapped in the zeolite were subsequently dissolved in hydrofluoric acid and recovered by liquid-liquid extraction using *n*-hexane. An Agilent 7890 gas chromatograph equipped with an Agilent HP5MS column (30 m, 320 µm, 0.25 µm film thickness) and a flame ionization detector (GC-FID) was used for identification and quantification of the *n*-alkanes, relative to external *n*-alkane standards (*n*-alkane mix *n*-C$_{21}$ - *n*-C$_{40}$, Supelco).
*n*-Alkane concentrations were calculated as the sum of C$_{25}$ to C$_{35}$, and are given in µg · g$^{-1}$ dry weight. Odd-over-even predominance (OEP) values (eq. 1) were determined following Hoefs et al. (2002). Low values (<5) indicate an enhanced state of degradation (Buggle et al., 2010; Zech et al., 2010). The Average Chain Length (ACL) (eq. 2) was calculated from the odd-numbered *n*-alkanes (Poynter et al., 1989).

$$\text{OEP} = \left( \frac{n\text{C}_{27} + n\text{C}_{29} + n\text{C}_{31} + n\text{C}_{33}}{n\text{C}_{26} + n\text{C}_{28} + n\text{C}_{30} + n\text{C}_{32}} \right) \quad \text{(eq. 1)}$$

$$\text{ACL} = \frac{27 \cdot n\text{C}_{27} + 29 \cdot n\text{C}_{29} + 31 \cdot n\text{C}_{31} + 33 \cdot n\text{C}_{33}}{n\text{C}_{27} + n\text{C}_{29} + n\text{C}_{31} + n\text{C}_{33}} \quad \text{(eq. 2)}$$





Compound-specific stable hydrogen isotope analyses of the $C_{31}$ and $C_{33}$ *n*-alkanes were carried out on an IsoPrime 100 IRMS,
coupled to an Agilent 7890A GC via a GC5 pyrolysis or combustion interface operating in pyrolysis modus with a MaxChrome
and silver wool packed reactor at 1050 °C. Samples were injected with a split–splitless injector. The GC was equipped with
30 m fused silica column (HP5-MS, 0.32 mm, 0.25 μm). Each sample was analysed in triplicate, except for single
measurements of three samples (VOV_915, VOV_128, VOV_111; numbers indicate sediment depth) due to insufficient
compound abundance. $\delta^2H_{n\text{-alkane}}$ was measured against calibrated $H_2$ reference gas and all values are reported in ‰ against
VSMOW. The precision was checked by co-analysing a standard alkane mixture (*n*-$C_{27}$, *n*-$C_{29}$, *n*-$C_{33}$) with known isotope
composition (Arndt Schimmelmann, University of Indiana), injected in duplicate every nine runs. All measurements were
corrected for drift and amount dependency, relative to the standard values in each sequence. Triplicates for the $C_{31}$ and $C_{33}$
alkanes had a standard deviation of <4.0 ‰, the analytical error for the standards was <1.7 ‰ ($n = 68$). The $H_3^+$ factor was
checked every two days and stayed stable at $4.40 \pm 0.03$ during measurements.
Compound-specific stable carbon isotope analyses of $C_{31}$ and $C_{33}$ *n*-alkanes were carried out on an IsoPrime 100 IRMS,
coupled to an Agilent 7890A GC via a GC5 pyrolysis or combustion interface operating in combustion modus with a CuO and
silver wool packed reactor at 850 °C. Samples were injected with a split–splitless injector. The GC was equipped with 30 m
fused silica column (HP5-MS, 0.32 mm, 0.25 μm). $\delta^{13}C_{n\text{-alkane}}$ values were calibrated against $CO_2$ reference gas of known
isotopic composition and all carbon isotope values are given in ‰ against VPDB. Triplicate injections were conducted for
each sample and measurement accuracy was controlled in the same way as for the $\delta^2H_{n\text{-alkane}}$ analyses. Triplicates for the $C_{31}$
and $C_{33}$ alkanes had a standard deviation of <0.2 ‰, the analytical error for the standards was <0.2 ‰ ($n = 83$).

## 4 Results

### 4.1 Lithology and Chronology

The sediment sequence has a length of 13 m and consists of three lithological units based on differences in sediment colour
(Fig. 2). Unit A from 13 to 4.51 m sediment depth consists of dark greyish material and is dominated by silt. Within this unit,
several thin greyish clayey and yellowish sandy layers are present. Unit B from 4.51 to 0.78 m sediment depth consists of
brownish to reddish silty material. In the lower parts of Unit B thin greyish clayey layers are observed. Unit C from 0.78 m
sediment depth to the top is dominated by dark brown silty material. Moreover, two layers (0.53 - 0.44; 0.31 - 0 m sediment
depth) can be separated through changes in colour and granulometric structure in Unit C (Fig. 2).


*[Figure 2]*

The chronology of the sediment record reveals a basal age of 8,440 $^{+200}/_{-250}$ cal BP (Fig. 2, Tab 2). In Unit A, $^{14}$C-ages of the
bulk samples are stratigraphically consistent, except for two samples at 11.19 and 5.66 m sediment depth, which are too young
for their stratigraphic position. The $^{14}$C-age ranges of a terrestrial organic macro particle and a reservoir corrected bulk



sediment sample at 9.63 and 9.61 m overlap. At 4.57 m sediment depth, two charcoal samples and a reservoir corrected bulk sediment sample distinctly overlap as well. Above the bulk sediment sample at 4.57 m sediment depth, four bulk sediment samples are too old for their stratigraphic position and amongst them not in stratigraphic order (Unit B and C). In contrast, [14]C-ages of compound-class *n*-alkane samples are distinctly younger and in stratigraphic order in Unit B and C (Tab 2; Fig. 2).

Consistent with the good dose recovery results the OSL sample produced a bright, rapidly decaying quartz OSL signal (reduced to ~2% of the initial signal within 1 second). None of the analysed aliquots exhibited recycling ratios or IR ratios beyond / below (respectively) 10% of unity or recuperation (zero dose signal) > 1% of the natural signal. The equivalent dose distribution however is overdispersed (40 ± 5%; Table 3), even after removal of one very high (125 Gy) outlier, despite the considerable signal averaging likely associated with the use of the fine sand fraction for analysis. In the context of the analysed

core, its likely antiquity, and the otherwise excellent performance of the sample implied by the dose recovery experiment and internal checks with in the SAR protocol, a parsimonious explanation of this broad distribution is the presence of unbleached or incompletely bleached grains. Indeed, the age obtained from the central age model equivalent dose estimate is 10 ± 1 ka, which is implausible given the stratigraphic position of the sample (Fig. 2). Application of a 3-component minimum age model (Galbraith et al., 1999) administered in the R package "Luminescence" (Burrow, 2019) produces a $D_e$ estimate of 20.3 ± 2.4

Gy and an age of 6.8 ± 0.8 ka (Table 3). While caution is required when applying this approach to multi-grain aliquots (especially those with considerable signal averaging as here), the minimum age estimate is much more concordant with the radiocarbon ages from this section of the core, and the broader age-depth model for the whole sequence (Fig. 2). In addition to that the MAM OSL-age is in stratigraphic order with the [14]C-ages of compound-class *n*-alkane and charcoal samples (Fig. 2).

*[Table 2]*

*[Table 3]*

**4.2 Geochemical and paleontological analyses**

Significant correlations (Pearson's r; α <0.05) were obtained for the log-normalised XRF scanning data for Br/Zr and Al/Zr compared to the quantitative elemental contents of Na and Al, respectively (r: Log (Br/Zr) / Na = 0.82; r Log (Al/Zr) / Al =

0.71; Fig. 3). Na concentrations range from 2,090 ± 110 to 10,390 ± 550 ppm over the sequence. Both Na contents and Br/Zr-ratios show highest values in Unit A (13 - 4.51 m sediment depth). Several minima in the Br/Zr-ratios derived from XRF-scanning data are not visible in the Na concentrations due to the higher sample resolution of the XRF-data (Fig. 3). Both Na concentration and Br/Zr-ratios distinctly decrease above 4.51 m sediment depth, but an increase is noticeable in Unit C (<0.78 m sediment depth) (Fig. 3). Al concentrations range from 39,300 ± 1,500 to 82,900 ± 3,100 ppm, and both Al contents

and Al/Zr-ratios show high values over large parts of the sediment record. As aforementioned, Al/Zr-ratios show minima in Unit A, which are not visible in the Al concentrations due to the higher sample resolution of the XRF-scanning data (Fig. 3). Noticeable are distinct low values of Al concentrations and Al/Zr-ratios from 4.51 to 2.91 m sediment depth and at 1.82 m sediment depth with an increasing trend to the top of the record (Fig. 3). TOC and N range from 0.16 to 3.94% and 0.03 to



0.48% respectively and show high values in the lower parts, strongly reduction above 4.51 m sediment depth and increase

thereafter again (Fig. 3). The C/N molar ratio ranges between 1.9 and 13.6 over the whole sequence with high values between 5.15 and 4.27 m sediment depth (Fig. 3). TIC ranges from 0 to 0.86% with noticeable peaks between 6.91 and 6.51, and at 4.83, 2.59 and 1.74 m sediment depth (Fig. 3). Ca contents show a similar pattern like TIC and ranges from $2{,}830 \pm 150$ to $28{,}900 \pm 1{,}600$ ppm (Fig. 3). BiSi concentrations range from $0.8 \pm 0.02$ to $2.8 \pm 0.08\%$ with low values in the lower parts of the record ($>6.99$ m sediment depth), except for high values between 10.98 and 10.10 m sediment depth

($1.6 \pm 0.05$ - $2.2 \pm 0.07\%$)(Fig. 3). They show high values for the upper part of the record $<6.91$ m sediment depth (Fig. 3). Apart from the selected mineral components of the XRD measurements displayed in Fig. 3, all samples have a predominance of quartz, feldspars and micas. Calcite follows the patterns of TIC and Ca and shows highest abundances between 11.46 and 10.18 as well as 3.87 and 2.19 m sediment depth (Fig. 3). The other sections have only medium to low proportions of calcite (Fig. 3). The presence of gypsum is restricted to depths between 9.62 and 9.14 m as well as 5.47 and 5.23 m (Fig. 3). However,

an absence of calcite and/or gypsum at the distinct peaks of TIC and Ca (e.g., 1.74 and 2.59 m sediment depth) is likely due to the lower sample resolution of the XRD measurements compared to the elemental data (Fig. 3). High proportions of halite are present in large parts of the sediment record ($>2.19$ m sediment depth; Fig. 3).

*[Figure 3]*


$n$-Alkane concentrations range from 0.22 to 5.23 µg·g$^{-1}$ with high values in Unit A and distinctly lower concentrations in Unit B and C (Fig. 4). All samples show a distinct odd over even predominance (4.0 - 15.5) and the ACL ranges from 29.9 to 31.4. Consequently, $C_{29}$, $C_{31}$ and $C_{33}$ are the predominant $n$-alkane chain-lengths in all samples (Fig. 4). $\delta^{13}C_{n\text{-alkane}}$ values range from $-28.10 \pm 0.16$ to $-22.72 \pm 0.06$‰ with $^{13}$C depleted values at the bottom of Unit A, followed by more $^{13}$C enriched

$\delta^{13}C_{n\text{-alkane}}$ values, which show a decreasing trend upwards (Fig. 4). Remarkable are three strongly $^{13}$C enriched $\delta^{13}C_{n\text{-alkane}}$ values in the upper parts of Unit A between 6.35 and 4.51 m sediment depth (Fig. 4). $\delta^{2}H_{n\text{-alkane}}$ values range from $-154.4 \pm 1.3$ to $-129.6 \pm 1.1$ ‰ with $^{2}$H enriched values at the bottom followed by a trend to $^{2}$H depleted values in Unit A. Unit B shows $^{2}$H enriched values at the bottom followed by $^{2}$H depleted values at the top and in Unit C (Fig. 4).

*[Figure 4]*

Paleontological analyses of the 33 macrofossil samples plus 23 microfossil samples revealed 56 snail specimens representing six species. In descending order of abundance these are *Turritella capensis* (Krauss, 1848), which is clearly dominant, the rare *Assiminea globulus (*Conolly, 1939), *Hydrobia* sp., *Natica tecta (*Anton, 1839), *Nassarius kraussianus* (Dunker, 1846) and a

fragment of an unidentified gastropod species (Fig. 5). The microfossil associations show a higher diversity (Fig. 5). The dominating foraminifer taxon is *Ammonia parkinsoniana* (d'Orbigny, 1839), followed by *Quinqueloculina* sp., and the rare *Trochammina inflata* (Montagu, 1808) and *Haynesina* sp. All other foraminifer taxa are represented by only one test each



(*Bolivina* sp., *Cribroelphidium articulatum* (d'Orbigny, 1839), *Spirillina* sp. and an unidentified trochamminid). The most abundant ostracod species are *Sulcostocythere knysnaensis* Benson and Maddocks, 1964 and juveniles of *Loxoconcha*

*parameridionalis*? Benson and Maddocks, 1964, rare ostracods are a myodocopid specimen and juvenile *Aglaiella* valves as well as the freshwater species *Sarscypridopsis aculeata* (Costa, 1847), *Cyprilla humilis* Sars, 1924 and an unidentifiable fragment of a larger species (Fig. 5). Microfossils identified on a group level only are ephippia of cladocerans, gyrogonites of charophytes, *Plumatella*-like bryozoan statoblasts, mollusc and insect fragments, fish bone remains, fruits and seeds as well as unidentifiable plant remains. Charcoal was found in large quantities in many samples.

Unit A1 shows a high diversity and contains most of the marine-brackish snails and high numbers of foraminifers (Fig. 5). The ostracod fauna is dominated by brackish water taxa. Shell and fish bone fragments are abundant. Unit A2 looks similar as Unit A1 but is generally more variable in abundances and diversity. Macrofossils, i.e., marine-brackish snails, become rare. Saltmarsh foraminifers, fruits and seeds occur for the first time (Fig. 5). Units B and C lack marine-brackish snails, brackish ostracods and saltmarsh foraminifers, and foraminifers in general are documented with a single test at the base of the zone

only. Freshwater ostracods occur for the first time and freshwater taxa in general dominate. Fragments of shells, plants, insects and charcoal disappear in Unit B and C (Fig. 5).

*[Figure 5]*

## 5 Discussion

**5.1 Chronostratigraphy**

Micropaleontological and (in)organic analyses show that three different depositional settings exist for the sediment sequence from Voëlvlei (Fig. 3, 5). In the lower part of the sequence (Unit A), the high concentrations of Na, TOC and N, high Br/Zr-ratios and marine/brackish gastropod species indicate a marine/brackish depositional setting from 13 to 4.51 m sediment depth (Fig. 2, 3, 5). Corresponding reservoir corrected bulk sediment, a terrestrial organic macro-particle and charcoal $^{14}$C-ages are

stratigraphically consistent and range from 8,510 $^{+280}/_{-200}$ to 6,310 $^{+450}/_{-210}$ cal BP indicating that sediments were rapidly deposited (Fig. 2). The distinct overlap of a bulk and the macro-particle age and bulk and charcoal ages confirm the reservoir correction and application of the Marine20 calibration curve (Heaton et al., 2020) for bulk samples in Unit A. The only exception are the bulk $^{14}$C-ages at 11.19 and 5.66 m sediment depth, which are too young for their stratigraphic position. This is probably due to an increased input of terrestrial organic carbon during this time indicated by high input of sand and reduced

Al contents and Al/Zr-ratios (Fig. 2, 3) which likely adds less $^{14}$C depleted material affecting the marine organic carbon stock and diluting the reservoir effect. As we cannot calculate the precise contribution of terrestrial organic carbon, these $^{14}$C-ages were excluded from the age model in a second modelling iteration (Heaton et al., 2020; Hogg et al., 2020) (Fig. 2).

At Voëlvlei, *n*-alkanes show a clear dominance of the longer chains $C_{31}$ and $C_{33}$, and thus are of terrestrial origin (Boom et al., 2014; Chambers et al., 2014; Strobel et al., 2020). Therefore, we calibrated them with the terrestrial SHCal20 calibration curve.





We note that minor contribution of the shorter chains ($<C_{25}$) to the dated compound-class *n*-alkane samples, which are synthesised by e.g., aquatic plants and possibly show a marine reservoir effect, would lead to too old $^{14}$C-ages when calibrated using SHCal20 calibration curve. The aforementioned dominance of long-chain *n*-alkanes however makes a terrestrial origin likely and the distinct overlap with marine calibrated and reservoir corrected bulk $^{14}$C-ages support the application of the SHCal20 calibration curve to the compound-class *n*-alkane samples (Fig. 2).

At 4.51 m sediment depth, a distinct shift in the depositional setting occurred with decreasing Na concentrations, Br/Zr-ratios, and TOC and N concentrations, pointing towards a change of the marine/brackish to a lacustrine environmental setting (Fig. 3, 5). Bulk $^{14}$C-ages are much older than their stratigraphic position and possibly show the deposition of degraded pre-aged sediments, possibly due to erosion of old organic carbon from deeper soil horizons within the catchment (Bliedtner et al., 2020; Douglas et al., 2018; Haas et al., 2020). The OSL-age is stratigraphically more consistent than the bulk $^{14}$C-ages. Since a MAM

was applied it is likely that this age reflects the timing of deposition, but the presence of a tailing of partially or unbleached sand grains also potentially implies an input from older sediments at this time, supporting the interpretation of the bulk $^{14}$C-age over-estimations. Compound-class *n*-alkane $^{14}$C-ages are stratigraphically consistent in Unit B and C. We therefore infer that leaf waxes derived from topsoils in the catchment and are rapidly deposited in the sediment archive (Bliedtner et al., 2020; Haas et al., 2017). Therefore, the incorporated climate signal should be close to that of the timing of deposition and *n*-alkane-

based proxies, i.e., $\delta^{13}C_{n\text{-alkane}}$ and $\delta^{13}C_{n\text{-alkane}}$, yield paleoenvironmental information that can also be interpreted robustly.

**5.2 Marine influence and lake development**

South Africa, especially the southern Cape coast, is known to have experienced distinct environmental changes related to relative sea level fluctuations during the Holocene (Cooper et al., 2018; Kirsten et al., 2018b; Wündsch et al., 2018; Wündsch et al., 2016a). Voëlvlei provides the possibility to contribute to our understanding of sea level changes during the Holocene

due to its present location at an elevation of 5 m a.s.l. and the high temporal resolution in the Early and Mid-Holocene. Inferred from lithological characteristics, paleoecological and elemental analyses, which indicate variable intrusion of marine waters, the record provides three eco-zones of which one can be subdivided in two subzones (Unit A1, A2, and B and C) (Fig. 6). During the period between 8,440 $^{+200}/_{-250}$ and 7,070 $^{+160}/_{-200}$ cal BP (Unit A1), the dominant gastropod *Turritella capensis* is a common modern species in sand in South African lagoons (Branch et al., 2010) preferring the mid-intertidal zone (Walters

and Griffiths, 1987) (Fig. 5, 6). The rarer gastropod *Assiminea globulus* is known to be very abundant on upper intertidal mudflats of South African estuaries today (Barnes, 2018), whereas the small gastropod *Hydrobia* prefers the upper salt marshes of South Africa (Branch et al., 2010). The single occurrences of *Natica tecta* and *Nassarius kraussianus* point to estuarine mudflats (Branch et al., 2010) (Fig. 5, 6). Abundant brackish water ostracods and foraminifers indicate permanent water cover at the coring site in this period. Summarising, the fauna reflects an estuarine and shallow subtidal environment close to

intertidal mudflats. The abundance of plant remains and fragments of shells and fish bones point to shallow water as well. Therefore, we assume the respective core depths reflect an elevation slightly below the past sea level, which is in line with the paleo-surface of lake Voëlvlei, i.e., 8.0 to 3.5 m below present sea level (b.s.l.) from 8,440 $^{+200}/_{-250}$ to 7,070 $^{+160}/_{-200}$ cal BP




(Fig. 6). The low diversity of foraminifers is typical for lower salinity conditions, the dominant small and unornamented *Ammonia* species, *Quinqueloculina* and *Haynesina* are consistent with this brackish water inference (Murray, 2006) (Fig. 5, 6). The same applies to the brackish water ostracod fauna, which are dominated by the estuarine species *Sulcostocythere knysnaensis* and *Loxoconcha parameridionalis* (Fürstenberg et al., 2017; Kirsten et al., 2018a) (Fig. 5, 6). The complete absence of open marine microfossils, especially planktic foraminifers and echinoderm fragments, indicates an inner estuarine position in the core locality without direct marine inflow, even under high-energy conditions. The few bryozoan statoblasts and gyrogonites of charophytes all derive from freshwater environments (Frenzel, 2019; Kirsten et al., 2018a) and point to an unconfined exchange with river water. The highest salinity for the entire core is probably reached at 7,090 $^{+170}/_{-200}$ cal BP when foraminifer abundance and diversity reach a maximum and a myodocopid ostracod was found (Fig. 5, 6). The coring position was ~3.65 m b.s.l. during this time (Fig. 6). Therefore, sea level was likely at the present height or slightly lower.

Between 7,070 $^{+170}/_{-200}$ and 6,420 $^{+130}/_{-140}$ cal BP (Unit A2), continued inner estuarine, brackish water conditions are indicated by a similar assemblage of taxa as observed in Unit A1. Intertidal gastropods typical for Unit A1, however, are now very rare, suggesting decreasing tidal influence and probably decreasing salinity (Fig. 5, 6). Salt marsh foraminifers occur only in Unit A2, albeit in low numbers (Fig. 6). This points to a very shallow water depth under marine/brackish conditions and a close shoreline (Strachan et al., 2017), which is in good agreement with the occurrence of fruits and seeds, only occurring in this unit, and abundant plant remains. However, the high variability in the abundance of many taxa indicates unstable conditions compared to Unit A1. Freshwater inflow is implied by floating bryozoan statoblasts and charophyte gyrogonites (Frenzel, 2019) (Fig. 5, 6). The respective paleo-surface of lake Voëlvlei was ~3.5 m b.s.l. to 0.5 m a.s.l. from 7,070 $^{+170}/_{-200}$ to 6,420 $^{+130}/_{-140}$ cal BP (Fig. 6) indicating a sea level higher than the present during this time.

Elemental (Br/Zr-ratios, Na contents) and mineralogical data support phases of marine water intrusions at lake Voëlvlei. Bromide salts are common in sea water, but occur in very low concentrations in freshwater systems (Song and Müller, 1993). In aqueous environments the dominant species is Br⁻, which substitutes the salt constituent chloride (Cl⁻) in the sea salt lattice during crystallization (Ullman, 1995). Halite (NaCl) also originates from marine waters, and thus Br, Na and halite can be used as indicators of marine water intrusion (Babel and Schreiber, 2014; Olsen et al., 2012; Wündsch et al., 2018). High values of the marine indicators from the elemental analysis (Na, Br/Zr-ratios) and the presence of halite in the sediments are in good agreement with our micropaleontological data indicating marine water intrusions in Units A1 and A2 (8,440 $^{+200}/_{-250}$ to 6,420 $^{+130}/_{-140}$ cal BP; Fig. 6). Furthermore, the occurrence of calcite in both units, as well as gypsum from 7,260 $^{+110}/_{-210}$ to 7,180 $^{+150}/_{-200}$ cal BP and 6,590 $^{+150}/_{-160}$ to 6,550 $^{+150}/_{-160}$ cal BP also points towards shallow water conditions as indicated by the micropaleontologically results (Fig. 6). Such shallow water conditions support bioproductivity (BiSi) increases in Unit A1 and A2 to its maximum at ~6,740 $^{+170}/_{-170}$ cal BP (Fig. 6). The elevation of the lake's floor rapidly increases from ~8 m b.s.l. at 8,440 $^{+200}/_{-250}$ (base of the VOV16 record) to ~0.5 m a.sl. at 6,420 $^{+130}/_{-140}$ cal BP (Unit A1 and A2; Fig. 6). Therefore, results of the Voëlvlei record underline a higher sea level compared to today until 6,420 $^{+130}/_{-140}$ cal BP and thus support a proposed local/regional relative sea level maximum of about +3.8 m a.s.l. (~7,600 – 5,800 cal BP) (Cooper et al., 2018) which occurred at the southern Cape.





The micropaleontological associations of Units B and C (6,420 $^{+130}/_{-140}$ cal BP until today) are completely different from those of Unit A1 and A2 (Fig. 5, 6). Many samples are barren of microfossils and freshwater taxa dominate other samples. One single test of the foraminifer *Ammonia* sp. found in the lowermost part of unit B and C is the only brackish water taxon (Murray,

2006) and probably reworked from older sediments where this species is very abundant (Fig. 5, 6). Therefore, we assume freshwater or athalassic conditions for samples with aquatic taxa. New to Unit B and C are the freshwater ostracod species *Sarscypridopsis aculeata* and ephippia of cladocerans, which are typical of non-permanent water bodies (Frenzel, 2019; Meisch, 2000). They likely reflect the transformation of Voëlvlei to a non-permanent lake and terrestrial habitat. The lower part of Unit B contains variable amounts of plant remains and charcoal indicating river transport. Their later disappearance

points to an isolation of the basin while silting up.

Decreasing elemental marine indicators (Na, Br/Zr-ratios) support reduced intrusion of marine waters from 6,420 $^{+130}/_{-140}$ cal BP    (Fig. 6).    Sedimentation    rates    markedly    decrease    while    the    occurrence    of    calcite (6,290 $^{+240}/_{-150}$ - 4,590 $^{+590}/_{-550}$ cal BP) further supports shallow water conditions in Units B and C (Fig. 6). While further silting up led to a location of the sediment surface above the present sea level (0 m a.s.l.) from 6,510 $^{+140}/_{-150}$ cal BP (Fig. 6) marine

water intrusion was likely absent from 4,300 $^{+490}/_{-570}$ cal BP when the sediment surface reached 3 m a.sl. This is based on considering a still elevated sea level of about +1 m a.sl. (5,300 – 4,200 cal BP) (Cooper et al., 2018) and a tidal range comparable to today (neap tide: 0.6 – 0.8 m, spring tide: 1.8 – 2 m) (Cooper, 2018, Rautenbach et al., 2019).

Overall, the results of this study only provide evidence of marine water intrusion in the Voëlvlei system rather than sea level index points and thus cannot be used to generate or corroborate an exact relative sea level curves (e.g., Compton, 2006; Cooper

et al., 2018). However, in local/regional comparison the results of this study are in line with the findings from Eilandvlei (Kirsten et al., 2018b; Wündsch et al., 2018) and Groenvlei (Wündsch et al., 2016a) which indicate a rising sea level during the Early Holocene and a high-stand during the Mid Holocene. They therefore, support the assumed sea level evolution at the southern Cape coast of South Africa (Kirsten et al., 2018b; Marker and Miller, 1993; Reddering, 1988; Wündsch et al., 2018; Wündsch et al., 2016a). In (supra-)regional comparison, these results are also in line with studies from the west coast of South

Africa (Baxter and Meadows, 1999; Carr et al., 2015; Kirsten et al., 2020) and Namibia (Compton, 2006), as reviewed in Cooper et al. (2018) showing a rapid sea level rise to a maximum of +3.8 m a.sl. (~7,600 – 5,800 cal BP), followed by decrease to +1 m a.sl. (5,300 – 4,200 cal BP) and a relatively constant sea level, comparable to the present, thereafter (Fig. 6).

*[Figure 6]*

**5.3 Paleoenvironmental and –climate evolution**

The main driver of the $\delta^2 H_{n\text{-}alkane}$ signal at Voëlvlei is the $\delta^2 H$ variability of the precipitation source (Strobel et al., 2020) although it has to be noted that variations in the vegetation composition, e.g., varying biosynthetic fractionation, evapotranspirative enrichment and water use efficiency possibly alter the $\delta^2 H_{n\text{-}alkane}$ signal (Hou et al., 2007; Sachse et al., 2012). However, modern reference material from topsoils suggest that Westerly-derived precipitation is $^2 H$ depleted and





Easterly-derived precipitation is $^2$H enriched pointing to the 'source effect' as dominant driver of the $\delta^2$H$_{n\text{-alkane}}$ signal at Voëlvlei (Braun et al., 2017; Harris et al., 2010; Strobel et al., 2020; Table 1). $\delta^2$H$_{n\text{-alkane}}$ values from Voëlvlei are moderate from 8,440 $^{+200}/_{-250}$ to 7,070 $^{+160}/_{-200}$ cal BP, which imply a year-round precipitation regime, i.e., contributions of both Westerly- and Easterly-derived precipitation (Fig. 7a).

Year-round precipitation is accompanied by $^{13}$C enriched $\delta^{13}$C$_{n\text{-alkane}}$ values between 8,440 $^{+200}/_{-250}$ and 7,070 $^{+160}/_{-200}$ cal BP,
likely indicating overall dry conditions for this period at Voëlvlei (Fig. 7f). These dry conditions potentially led to a sparse vegetation cover and runoff (induced by occasional events) which likely carries extremely variable grain sizes and amounts of allochthonous input (Al concentration, Al/Zr-ratios) (Fig. 7h). Minima in Al concentrations and Al/Zr-ratios are assumed to be caused by layers consisting nearly exclusively of sand (SiO$_2$) and thus also imply high allochthonous inputs (Fig. 2, 7h). Similarly at Vankervelsvlei, input of reworked soil material during this time is likely a result of low vegetation cover in the
catchment (Strobel et al., 2019). This interpretation of dry conditions between 8,440 $^{+200}/_{-250}$ and 7,070 $^{+160}/_{-200}$ cal BP is also in line with the findings from Eilandvlei, where low Afrotemperate forest (AFT) pollen percentages are interpreted as overall dry conditions between ~8,500 and 7,000 cal BP (Quick et al., 2018) (Fig. 7e) and a study at Seweweekspoort, where $^{15}$N depleted $\delta^{15}$N values are also interpreted to indicate moderately dry conditions (Chase et al., 2017) (Fig. 7c) resulting in a consistent moisture signal in the YRZ during this time.


*[Figure 7]*

At 7,020 $^{+170}/_{-200}$ cal BP, overall moister conditions are indicated by a distinct shift to $^{13}$C depleted $\delta^{13}$C$_{n\text{-alkane}}$ values and high allochthonous input (Al content, Al/Zr-ratios), which lasts until present. Although, there is a decrease in allochthonous input
from 6,420 $^{+130}/_{-140}$ to 6,080 $^{+320}/_{-240}$ cal BP, and at 4,480 $^{+580}/_{-540}$ and 3,530 $^{+500}/_{-470}$ cal BP (Fig. 7h). The latter is accompanied by distinctly $^{13}$C depleted $\delta^{13}$C values (Fig. 7f) whereas the earlier one only shows slightly $^{13}$C depleted values. This potentially indicates a denser vegetation cover, reducing runoff-induced sediment input, which is supported by the presence of freshwater taxa during this time (Fig. 5). However, drier conditions inferred from reduced allochthonous input could also be possible which might cause drought-induced changes of the vegetation composition, i.e., higher proportion of plants using CAM
metabolism. These could possibly reduce drought-stress and therefore the influence of water use efficiency to $^{13}$C enrichment in the $\delta^{13}$C$_{n\text{-alkane}}$ values (Diefendorf and Freimuth, 2017). Therefore, the climatic deviations at 4,480 $^{+580}/_{-540}$ and 3,530 $^{+500}/_{-470}$ cal BP interrupting the trend towards moister conditions cannot be ultimately attributed to even moister or drier conditions. Noticeable are three data points at 6,670 $^{+170}/_{-170}$, 6,560 $^{+150}/_{-160}$ and 6,440 $^{+130}/_{-140}$ cal BP showing strongly $^{13}$C enriched $\delta^{13}$C$_{n\text{-alkane}}$ values which exceed all other measurements and are accompanied by distinctly $^2$H depleted $\delta^2$H$_{n\text{-alkane}}$
values (grey dots in Fig. 7a, e). For those three samples, the climate signal is possibly overprinted by local effects of plants that grow at shallower water depths and/or at the shoreline of the lake. While $^{13}$C enriched $\delta^{13}$C$_{n\text{-alkane}}$ values point to the presence of plants using C4 and/or CAM metabolism (Boom et al., 2014; Carr et al., 2015; Diefendorf and Freimuth, 2017), the $\delta^2$H$_{n\text{-alkane}}$ signal with $^2$H depleted values can be biased by the photosynthetic mode and also salinity (Aichner et al., 2017;



Feakins and Sessions, 2010; Ladd and Sachs, 2012; Sachse et al., 2012). Therefore, disentangling the regional climate and

local overprinting is challenging for the three aforementioned data-points. However, since allochthonous input is high (Al concentration, Al/Zr-ratios; Fig. 7h), we hypothesize, overall moist conditions since 7,020 $^{+170}/_{-200}$ cal BP.

Moister conditions are accompanied by $^2$H depleted $\delta^2$H$_{n\text{-alkane}}$ values from 7,020 $^{+170}/_{-200}$ to 6,420 $^{+130}/_{-140}$ cal BP which are likely indicative for a dominance of Westerly-derived precipitation (Fig. 7a). Contemporary to the aforementioned reduction of the allochthonous input, there is also a marked shift to $^2$H enriched $\delta^2$H$_{n\text{-alkane}}$ values at 6,420 $^{+130}/_{-140}$ implying an increased

contribution of Easterly-derived precipitation (Fig. 7a). Afterwards, $\delta^2$H$_{n\text{-alkane}}$ values are variable, but have an overall trend to $^2$H depleted $\delta^2$H$_{n\text{-alkane}}$ values lasting until 2,060 $^{+140}/_{-200}$ cal BP which is likely indicative for a return to a dominance of Westerly-derived precipitation (Fig. 7a). However, this period is interrupted by a short phase of somewhat $^2$H depleted values from 5,560 $^{+430}/_{-470}$ to 5,150 $^{+340}/_{-330}$, indicating more Westerly-derived precipitation (Fig. 7a). From 2,060 $^{+140}/_{-200}$ cal BP until present, $^2$H enriched $\delta^2$H$_{n\text{-alkane}}$ values imply an increased contribution of Easterly-derived precipitation (Fig. 7a), which occurs

within a trend towards moister conditions ($^{13}$C depleted $\delta^{13}$C$_{n\text{-alkane}}$) (Fig. 7f).

Moist conditions between 7,000 and 4,700 cal BP are in line with studies from the central southern Cape coast (Quick et al., 2018; Strobel et al., 2019; Wündsch et al., 2018). However, beside increased moisture, also an increased wind driven evapotranspiration was reconstructed at Vankervelsvlei (Strobel et al., 2019). The latter is also possibly related to an increased Westerly wind belt and thus in line with the findings of this study showing intensified Westerly-derived precipitation at

Voëlvlei (Fig. 7d, e). At ~4,700 cal BP, climatic spikes were also detected in numerous studies located in the YRZ. In the Wilderness area at the southern Cape coast, dry conditions but low wind-driven evapotranspiration were reconstructed during this time (Quick et al., 2018; Strobel et al., 2019; Wündsch et al., 2018) (Fig. 7d, e). After 4,700 cal BP, the overall trend in increasing moisture availability at the central southern Cape coast is again in line with our findings at Voëlvlei (du Plessis et al., 2020; Quick et al., 2018; Strobel et al., 2019; Wündsch et al., 2018). $\delta^2$H$_{n\text{-alkane}}$ and $\delta^{13}$C$_{n\text{-alkane}}$ from marine sediments

recovered off the mouth of the Gouritz River (GeoB18308-2; Fig. 1) show a very similar pattern to Voëlvlei over the past 4,000 cal BP (Hahn et al., 2017) (Fig. 7). Those findings thus support our findings from Voëlvlei providing a consistent regional picture for the southern Cape coast.

At Seweeekspoort, ~100 km inland, $\delta^{15}$N values follow the pattern of $\delta^2$H$_{n\text{-alkane}}$ from Voëlvlei (Fig. 7c). There, climate is thought to be dry from ~7,000 to ~4,700 cal BP and ~3,000 to ~1,000 cal BP due reduced Easterly-related precipitation (Chase

et al., 2017) (Fig. 7c). Climatic spikes to moister conditions at Seweeekspoort contradict the aforementioned results from the wilderness area during this time, but are well reflected in spikes in the $\delta^2$H$_{n\text{-alkane}}$ record from Voëlvlei towards $^2$H enriched $\delta^2$H$_{n\text{-alkane}}$ values indicating increased Easterly-related precipitation contribution (Fig. 7a, c). In this context, previous studies hypothesise that the non-dominant component of the temperate (Westerly)/tropical (Easterly) dynamic determines hydro-climatic variability in southern Africa (e.g., Chase et al., 2017; Chase et al., 2015). More precisely, this means that although

Easterly (Westerly)-derived precipitation is relatively low in the western (eastern) parts of the YRZ, variability in the overall moisture signal is thought to strongly depend on moisture contribution from this minor component (e.g., Chase et al., 2017; Chase et al., 2015). The $\delta^2$H$_{n\text{-alkane}}$ values of this study imply increased Westerly-derived precipitation during drier phases at





Seweweekspoot and increased Easterly-derived precipitation during moister phases at Seweweekspoort (Fig. 7c), at a time generally characterised by a trend towards moister conditions at Voëlvlei (Fig. 7f). Moist conditions between 7,000 and 5,500 cal BP were also found in the north-south expanded Cape Fold Mountains at the western Cape (Chase et al., 2019 and references therein), indicating an overall moist period for south-western South Africa during this time which is perfectly in line with the increased Westerly precipitation contribution at Voëlvlei.

However, an opposite pattern is proposed to exist between sites located inland of the Cape Fold mountains (e.g., Seweweekspoort, Katbakkies Pass) and coastal sites from the southern Cape coast (e.g., Eilandvlei) (Chase and Quick, 2018; Quick et al., 2018). There are still discrepancies due to the comparison of different proxies in these studies compared to those from the southern Cape coast. Comparing our $\delta^2H_{n\text{-alkane}}$-record from Voëlvlei, which is representative for fluctuations in the major moisture sources, with the $\delta^{15}N$ from Seweweekspoort (Chase et al., 2017), in detail many similarities are apparent confirming the assumption that hydrological variations at Seweweekspoort are driven by variations in Easterly-derived precipitation. In contrast, at the South Coast of South Africa including Voëlvlei the overall moisture evolution is not restricted to one moisture source and thus driven by a combination of both Westerly- and Easterly-derived precipitation.

The topmost sediments of Voëlvlei mainly consist of two fluvial flooding events (at 830 $^{+110}/_{-110}$ and 10 $^{+10}/_{-10}$ cal BP) which deposited large amounts of sediment (~15 cm and ~30 cm, respectively; Fig. 2; 7). Considering the dating error, the most recent flooding event is likely to be associated with the so called 'Laingsburg flood' from January 1981 (Damm and Hagedorn, 2010 and references therein). In marine sediments recovered off the mouth of the Gouritz River (GeoB18308-2; Fig. 1) major flooding events were detected between 650 and 300 cal BP and thus support our findings (Hahn et al., 2017).

## Conclusions

Our multi-proxy record from lake Voëlvlei provides new insights in the sea level and paleoclimate history of the past 8.5 ka at the southern Cape Coast. Various dating approaches including OSL and $^{14}C$ of bulk TOC, organic macrofossils, charcoal and $n$-alkanes, were applied to the sediments to establish a robust chronology. Three temporal eco-zones can be characterised for Voëlvlei. These are related to sea level variations and are in line with the results of regional investigations in implying a subtidal marine-brackish followed by a intratidal brackish from 8,440 $^{+200}/_{-250}$ to 6,420 $^{+130}/_{-140}$ cal BP, and a freshwater to terrestrial system due to silting up of Voëlvlei, thereafter. In terms of climate, dry conditions accompanied by year-round precipitation contribution prevail from 8,440 $^{+200}/_{-250}$ to 7,070 $^{+160}/_{-200}$ cal BP followed by moister conditions and more Westerly-derived precipitation contributions from 7,070 $^{+160}/_{-200}$ to 6,420 $^{+130}/_{-140}$ cal BP followed by a distinct shift to an Easterly-dominance at 6,420 cal BP. An overall trend to a Westerly- lasting until 2,060 cal BP is followed by a trend towards an Easterlies-dominance, but both phases show several climatic spikes.

Our results are in good agreement with previous studies with regard to sea level changes on the western and southern coast of South Africa. The paleohydrological evolution of Voëlvlei is comparable to previous investigations from the Southern Cape coast during the past 8.5 ka, demonstrating an interplay of both Westerly- and Easterly-derived precipitation contribution. In



580 contrast, hydrological variations at Seweweekspoort, located within the interior of central southern South Africa, a region hypothesised to show distinct hydroclimatic trends (relative to the coastal zones) through the Holocene, shows many similarities to the Voëlvlei $\delta^2H_{n\text{-alkane}}$ record, indicating that moisture is driven by variations in Easterly-derived precipitation there. Thus, the Voëlvlei $\delta^2H_{n\text{-alkane}}$ record provides valuable insights in the source of precipitation at the Southern Cape coast during the past 8.5 ka.

## Acknowledgements

This study was funded by the German Research Foundation (DFG) (HA 5089/11-1; ZE 860/6-1) and the German Federal Ministry of Education and Research (BMBF) within the collaborative projects "Regional Archives for Integrated Investigations" (RAiN) (grant no.: 03G0862B) and TRACES (Tracing Human and Climate impacts in South Africa) (grant no.: 03F0798A) which are embedded in the international research programme SPACES (Science Partnership for the Assessment of Complex Earth System Processes). PS gratefully acknowledges the support by a fellowship from the state of Thuringia (Landesgraduiertenstipendium). Particularly acknowledged are C. Berndt, N. Blaubach, T. Eggert, C. Gregori, P. Rauh, M. Steinich and M. Wagner for assistance in the lab. G. Daut is thanked for supporting the XRD measurements. N. du Plessis, J. Baade, S. Hess, C. Gregori, R. Mäusbacher and M.E. Meadows are thanked for their fieldwork contributions and constructive discussions. The participants of the 2019 course on Paleoenvironmental Analysis at Friedrich Schiller University Jena supported us in micropaleontological sample preparation and picking of microfossils for an initial batch of subsamples.

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



**Figures**



**Figure 1: A) Simplified map of Africa. The red box highlights the studied area. B) Location of lake Voëlvlei and studies mentioned in the text. Additionally, the circumpolar Westerlies, the tropical Easterlies, the Agulhas Current (AC) and the Benguela Current (BC) are depicted. C) Voëlvlei, its catchment, the coring position as well as a paleo-sea level highstand (+5 m) and the recent shoreline (Data source: Rainfall seasonality: Worldclim 2 dataset (Fick and Hijmans, 2017); Circulation systems after Chase and Meadows (2007), DEM: SRTM 1 arc-second (~30 m)).**




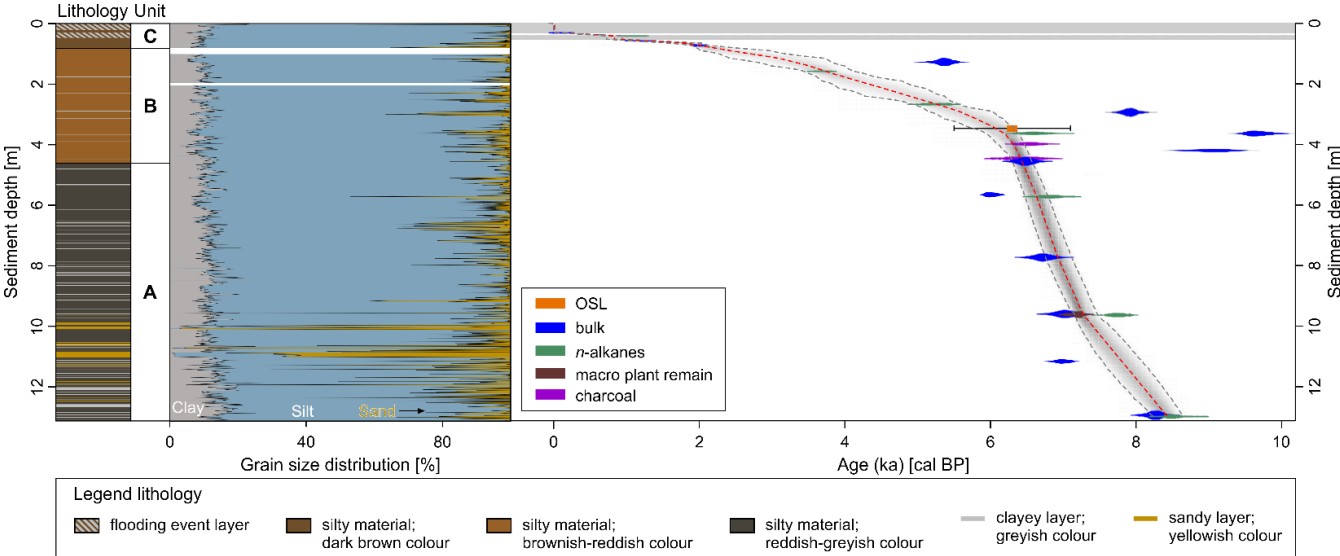

**Figure 2: left: Lithology and grain size distribution (clay, silt, sand) of the VOV16 record from Voëlvlei. Right: Age-depth model of the sediment record from Voëlvlei. Calibrated radiocarbon ages are displayed as probability density functions of the 2σ distributions (blue: bulk sediment; green: compound-class n-alkane samples; brown: macro plant remain). Calibration and age-depth modelling was carried out using the R software package Bacon 2.4.3 (Blaauw and Christen, 2011).The logo of Copernicus Publications.**






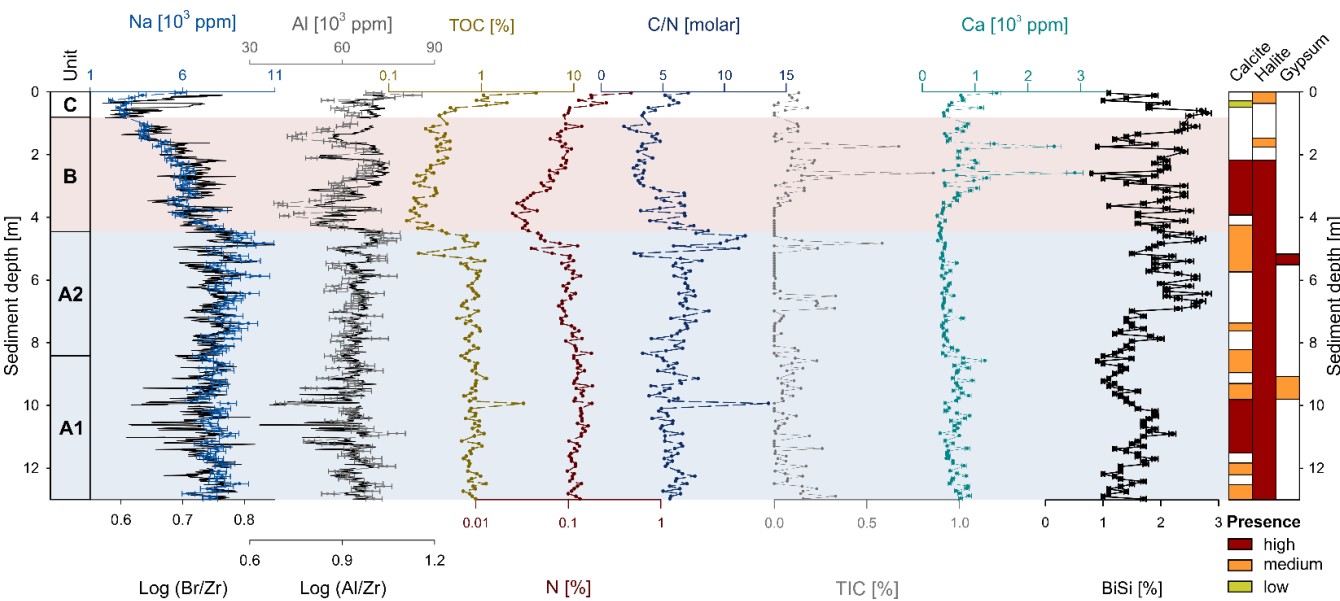

**Figure 3: Lithological units, Sodium (Na) content and Log$_{10}$ ratio of Bromine (Br) and Zirconium (Zr), Aluminium (Al) and Log$_{10}$ ratio of Al and Zr, contents of Total Organic Carbon (TOC), total Nitrogen (N), molar C/N-ratio, Total Inorganic Carbon (TIC), Calcium (Ca) and Biogenic Silica (BiSi) derived from the sediment core VOV16 from Voëlvlei. The presence of selected mineral components is also depicted.**





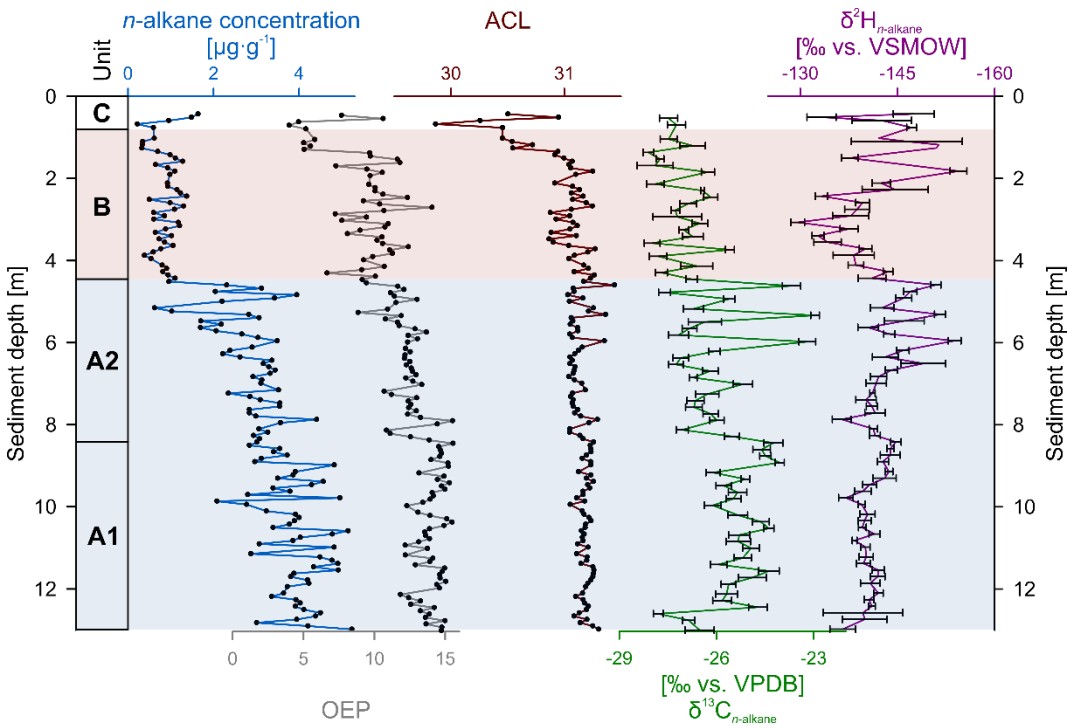

**Figure 4: Concentration, odd-over-even predominance (OEP) and average chain length (ACL) of leaf wax *n*-alkanes and their stable isotopic composition for hydrogen (δ²H*n*-alkane) and carbon (δ¹³C*n*-alkane) of the Voëlvlei sediment record.**



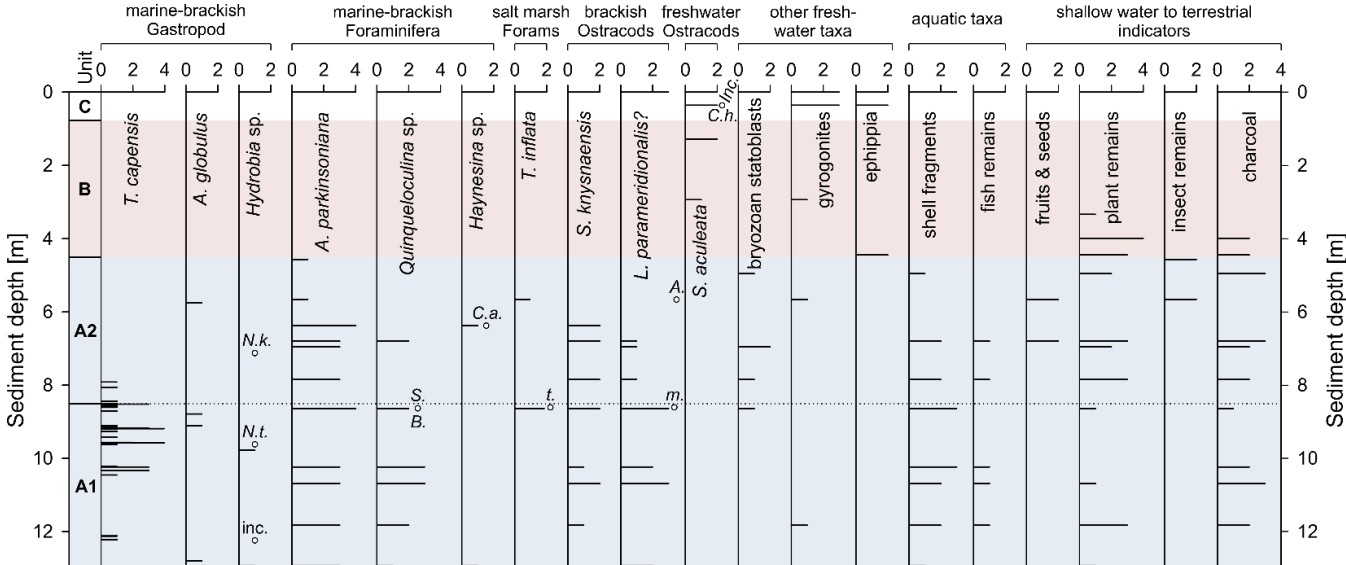

**Figure 5: Distribution of micropaleontological taxa and charcoal in core VOV16. Units are based on distribution of micropaleontological taxa and lithological characteristics. Additional macrofossils were picked from the lowest part of the core where they are relatively abundant. All abundances are given semi-quantitatively (0 – absent, 1 – rare, 2 – common, 3 – abundant, 4 – very abundant). Single occurrences of taxa are indicated by empty circles and abbreviated name (Nassarius kraussianus, Natica tecta, Bolivina sp., Spirillina sp., Cribroelphidium articulatum, trochamminid foraminifer, Aglaiella, myodocopid ostracod, Cyprilla humilis).**






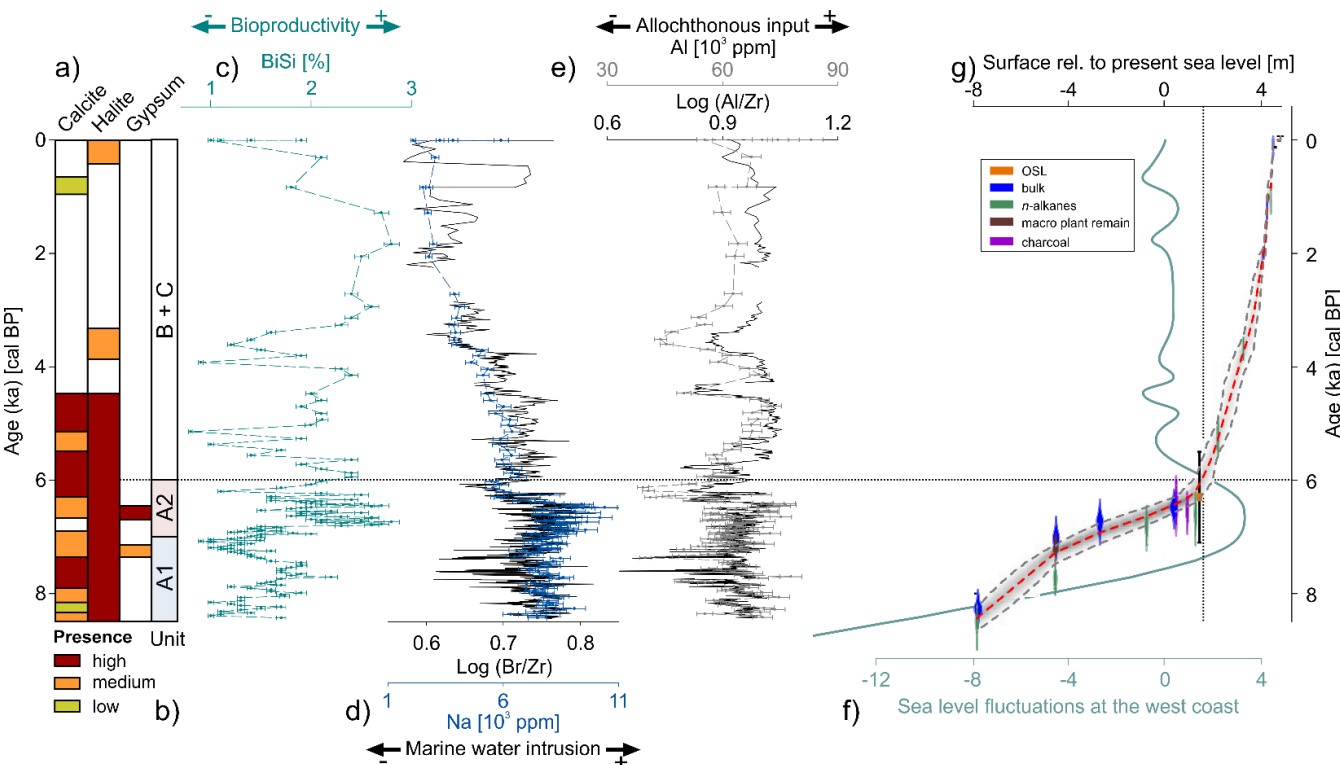

**Figure 6: a)** Occurrence of selected mineral components in the VOV16 sediment record, **b)** lithological and paleoecological units (A1 – subtidal marine-brackish, A2 – intratidal brackish, B+C freshwater to terrestrial), **c)** Biogenic Silica (BiSi) content, **d)** Na content and Log (Br/Zr)-ratios, **e)** Al content and Log (Al/Zr)-ratios from Voëlvlei. **f)** Sea level curve for the west coast of South Africa (Compton, 2006; Cooper et al., 2018) and **g)** the lake floor of Voëlvlei relative to the present sea level.





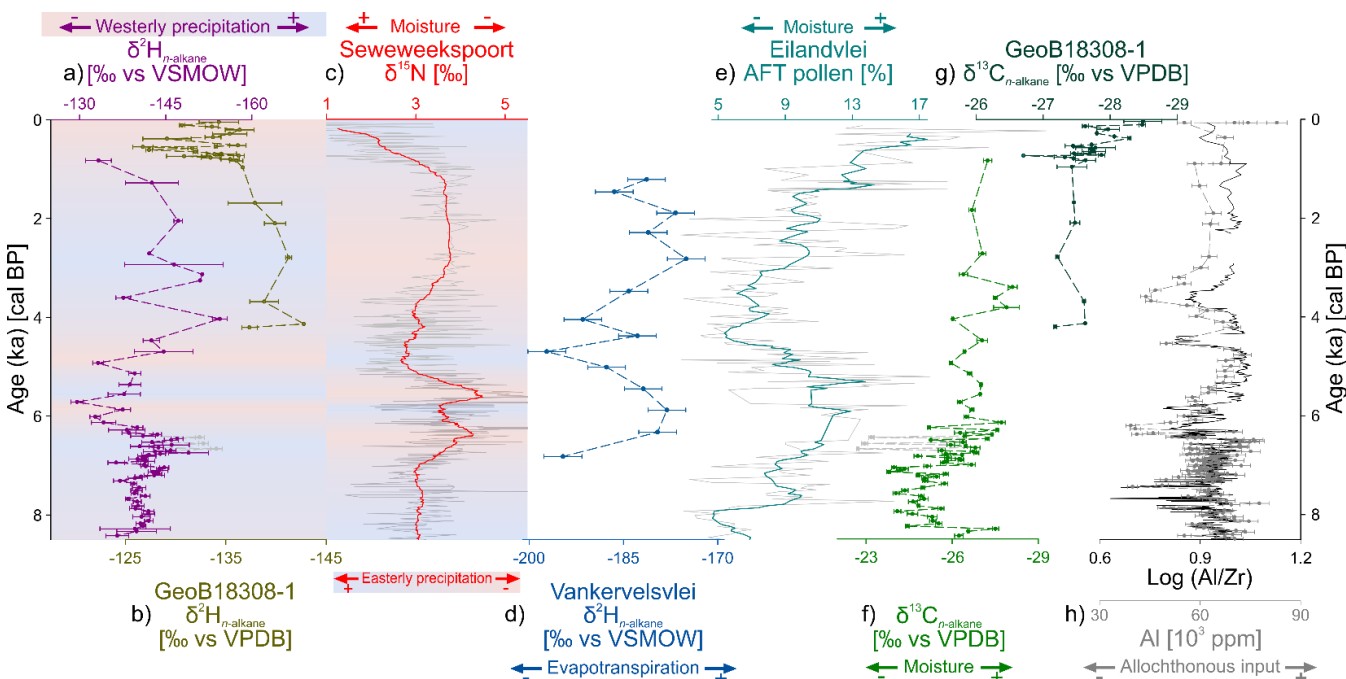

**Figure 7: a)** $\delta^2H_{n\text{-alkane}}$ **from Voëlvlei compared to b)** $\delta^2H_{n\text{-alkane}}$ **from GeoB18308-1 (Hahn et al., 2017), c)** $\delta^{15}N$ **from Seweweekspoort**
**(Chase et al., 2017), d)** $\delta^2H_{n\text{-alkane}}$ **from Vankervelsvlei (Strobel et al., 2019), e) Afrotemperate forest pollen (AFT) from Eilandvlei**
**(Quick et al., 2018), f)** $\delta^{13}C_{n\text{-alkane}}$ **from Voëlvlei, g)** $\delta^{13}C_{n\text{-alkane}}$ **from GeoB18308-1 (Hahn et al., 2017), and h) Aluminium (Al) content**
**and Log Al/Zr-ratio from Voëlvlei. The locations of the studies are depicted in Figure 1.**



**Tables**

**Table 1: Modelled isotopic hydrogen composition of precipitation at Voëlvlei (Lat: 34.013° S; Lon: 22.904° E; Elevation: 5 m) (Bowen, 2018; Bowen et al., 2005).**

|  | Jan | Feb | Mar | Apr | May | Jun | Jul | Aug | Sep | Oct | Nov | Dec |
|---|---|---|---|---|---|---|---|---|---|---|---|---|
| $\delta^2 H_p$ [‰ vs. V-SMOW] | 6 | 4 | 3 | -12 | -21 | -26 | -29 | -33 | -23 | -16 | -12 | -4 |





**Table 2: Conventional radiocarbon ages as well as 2σ calibrated age ranges and median calibrated ages (Calib 8.2) (Stuiver et al.,**
**2020) using the SHCal20 and Marine20 calibration curve (Heaton et al., 2020; Hogg et al., 2020) of dated bulk sediment, n-alkanes and organic macro particle (OMP) samples from the VOV16 record. Samples from the marine and brackish parts of the record were reservoir corrected using a ΔR of 134 ± 38 $^{14}$C yrs after Wündsch et al. (2016b). Too old and too young samples are highlighted by [*] and [#], respectively.**

| Sediment depth [m] | Lab ID | 1σ conventional $^{14}$C age [BP] | Dated material | Median cal age and 2σ error [cal BP] | Calibration |
|---|---|---|---|---|---|
| 0.29 | Poz-94013 | 660 ± 30 | bulk | 610 $^{+75}$/$_{-25}$ | 1 |
| 0.32 | BE-9770.1.1 | -85 ± 120 | bulk | 100 $^{+85}$/$_{-85}$ | 1 |
| 0.44 | BE-12027.1.1 | 1,250 ± 140 | n-alkanes | 1,110 $^{+250}$/$_{-130}$ | 1 |
| 0.51 | BE-9772.1.1 | 1,855 ± 120 | bulk | 1,740 $^{+270}$/$_{-270}$ | 1 |
| 0.59 | BE-9772.1.1 | 1,360 ± 120 | bulk | 1,210 $^{+470}$/$_{-250}$ | 1 |
| 0.75 | Poz-96323 | 2,080 ± 35 | bulk | 2,000 $^{+170}$/$_{-80}$ | 1 |
| 1.29 | Poz-94014 | 5,330 ± 50 | bulk | 5,370 $^{+410}$/$_{-210}$ | 2 [*] |
| 1.59 | BE-12026.1.1 | 3,470 ± 90 | n-alkanes | 3,690 $^{+455}$/$_{-240}$ | 1 |
| 2.68 | BE-12025.1.1 | 4,630 ± 100 | n-alkanes | 5,280 $^{+520}$/$_{-310}$ | 1 |
| 2.95 | Poz-96324 | 7,780 ± 50 | bulk | 7,920 $^{+380}$/$_{-190}$ | 2 [*] |
| 3.64 | BE-12024.1.1 | 5,830 ± 130 | n-alkanes | 6,600 $^{+590}$/$_{-290}$ | 1 |
| 3.65 | Poz-94016 | 9,250 ± 80 | bulk | 9,660 $^{+550}$/$_{-250}$ | 2 [*] |
| 3.99 | BE-13598.1.1 | 5,780 ± 120 | charcoal | 6,550 $^{+500}$/$_{-250}$ | 1 |
| 4.21 | Poz-98909 | 8,750 ± 170 | bulk | 9,060 $^{+900}$/$_{-480}$ | 2 [*] |
| 4.57 | Poz-98910 | 6,380 ± 40 | bulk | 6,490 $^{+370}$/$_{-190}$ | 2 |
| 4.57 | BE-13597.1.1 | 5,550 ± 110 | charcoal | 6,310 $^{+450}$/$_{-210}$ | 1 |
| 4.57 | BE-13596.1.1 | 5,800 ± 120 | charcoal | 6,570 $^{+500}$/$_{-270}$ | 1 |
| 5.66 | Poz-94017 | 5,740 ± 40 | bulk | 5,800 $^{+370}$/$_{-200}$ | 2 [#] |
| 5.73 | BE-12023.1.1 | 6,020 ± 100 | n-alkanes | 6,830 $^{+470}$/$_{-230}$ | 1 |
| 7.74 | Poz-94018 | 6,590 ± 50 | bulk | 6,720 $^{+420}$/$_{-210}$ | 2 |
| 9.61 | Poz-94021 | 6,850 ± 50 | bulk | 7,020 $^{+410}$/$_{-210}$ | 2 |
| 9.63 | Poz-94020 | 6,310 ± 50 | OMP | 7,200 $^{+160}$/$_{-50}$ | 1 |
| 9.64 | BE-12022.1.1 | 6,940 ± 110 | n-alkanes | 7,750 $^{+370}$/$_{-180}$ | 1 |
| 11.19 | Poz-94022 | 6,620 ± 50 | bulk | 6,760 $^{+420}$/$_{-210}$ | 2 [#] |
| 12.95 | Poz-94023 | 8,110 ± 50 | bulk | 8,270 $^{+370}$/$_{-210}$ | 2 |
| 12.99 | BE-12021.1.1 | 7,750 ± 110 | n-alkanes | 8,510 $^{+460}$/$_{-190}$ | 1 |

*1 SHCal20, 2 Marine20 and reservoir corrected using a ΔR of 134 ± 38 $^{14}$C yrs after Wündsch et al. (2016b)*






**Table 3: Equivalent dose measurements on sample VOV16-1. 24 aliquots were measured (a 25th with a De of ~125 Gy was excluded prior to analysis).**

| Average sediment depth [m] | Dose rate (Gy ka$^{-1}$) | CAM $D_e$ (Gy) | CAM OD [%] | MAM De (Gy) | CAM age (ka) | MAM age (ka) |
|---|---|---|---|---|---|---|
| 3.50 | $3.01 \pm 0.12$ | $31.0 \pm 2.8$ | $40 \pm 6$ | $20.3 \pm 2.3$ | $10.3 \pm 1.0$ | $6.8 \pm 0.8$ |