# Peer review of "Holocene sea level and environmental change at the southern Cape - an 8.5 kyr multi-proxy paleoclimate record from lake Voëlvlei, South Africa"

_Climate of the Past, 2020_

## Referee Comment (RC1) · Hayley Cawthra (Referee) · 25 Dec 2020

Paul Strobel and colleagues have presented a paper that documents the results of a study in the southern Cape of South Africa. That these authors are well experienced in working on Holocene records from this region shows in their carefully-compiled dataset and detailed interpretations that cover the state of knowledge on this coast and its hinterland. They present a multi-proxy investigation on a sediment core, and as well as radiocarbon dating, apply OSL to the record to provide two methods of geochronology.

This site is well positioned to compare a coastal plain to an offshore core record from the adjacent shelf, linked to the same Gourits River catchment.

[Figure]

Overall, I suggest only minor revisions to this well-constructed paper. One broad comment I have, however, is concerning modern-day drought and discussions of past drought events. By the end of the paper, there is a clear history of the relative changes in moisture during the period under consideration, but is it possible to say more explicitly in the text just 'how' dry or how moist the climate may have been, compared to a benchmark of today for example?

Specific suggestions for in-text edits:

Line 18: sea-level changes

Lines 19, 23, 78: I am not happy with this term 'intermittent' as it is not clear what you mean. Please define exactly what this refers to in the introduction.

Line 29: what is a climatic spike?

Line 30: major, not mayor

Line 31-32: the last sentence of the abstract promises evidence of changing moisture, but the results highlighted in the abstract talk to shifts between easterlies and westerlies. Readers who are not familiar with the area and its climate regime will not be able to infer what that means, so please be more clear with giving a brief summary of moisture fluctuation in the abstract.

Lines 34-35: this correlation to modern needs for studying past climate needs to be better integrated by the end of the study, as this was the only specific reference I noted to the present conditions. A short section in the discussion, even if only a few lines, to tie back to this 'why we learn about the past' is necessary to loop back to the departure point of the paper.

Line 39: currents

Line 79: delete 'therefore'

Line 87: the reference for the SRTM dataset is Jarvis, A., H.I. Reuter, A. Nelson, E.

Guevara, 2008, Hole-filled SRTM for the globe Version 4, available from the CGIAR-CSI SRTM 90m Database (http://srtm.csi.cgiar.org)

Line 90: what is the name of the ephemeral river? Is it the Buffels River?

Line 119: write out 'Twelve' to start the sentence

Line 269: you write that the different sediment units were differentiated on colour, but it appears that grain size was considered too?

Line 314: 'strongly reduction' needs rephrasing

Line 317: pattern to TIC

Line 335: 'Remarkable are three strongly' needs rephrasing. I suggest saying 'Notably, . . .'

Line 401: marine water

Line 405: delete 'very'

Line 464: sea-level curves

Line 467: sea-level evolution

Line 577: sea-level changes

A few linguistic peculiarities to address:

Line 23: rephrase to: silting up has resulted in an intermittent freshwater lake

Line 37: hitherto is not correct in this context

Line 310: rather use 'mentioned previously' instead of 'aforementioned'

There are a few references missing from the list:

Line 344: Conolly, 1939

Line 346: d'Orbigny, 1839

Line 347: Montagu, 1808

Line 351: Costa, 1847

Line 379: Chambers et al., 2014

Line 462: Rautenbach et al., 2019

All figures are good quality and all contribute meaningful data and information.

I look forward to seeing this manuscript published.

Kind regards

---

## Referee Comment (RC2) · Anonymous Referee #2 · 18 Feb 2021

Dear editors of Climate of the Past

I apologise for the length of time this took to reach my decision, as you will note below, it is complicated.

This paper highlights state of the art analyses (mostly inaccessible to African scientists due to budgetary constraints) applied to a core from the south coast of South Africa. The story is relatively clear, though better attention can be paid to the geomorphic functioning of the system relative to sea-level change. This is because a core is merely a one dimensional facet in a system that responds dynamically to changing sea level and which inevitably manifests sedimentologically and geochemically via multiple feedback loops between the system geomorphology and core constituents. This is often ignored in these systems with the assumption that they are mostly steady state and thus a climate or sea level archive. Better discussion of geomorphic influences needs to be made. Without this, I cannot completely trust the archive status of the core and results thereafter.

On a more serious topic, and something I have wrestled with for some time now, I feel I can no longer be complicit in the denudation and slow drain of research resources and ideas from Africa (and other developing nations) by the processes of scientific colonialism. The paper has not a single local author in the long team of people involved, with merely an acknowledgement to South African field workers. It remains unclear what all authors did, and why indeed a multinational and internationally authored paper and project funded by the BMBF is permitted to be submitted without inclusion of local knowledge and authors. Attached to this review, and available from the below links are several articles, tweets and papers that highlight a serious issue surrounding scientific imperialism and the continued exclusion of African and developing world scientists by better funded and resourced first world teams.

By excluding local scientists who I presume were at some point included in the proposal in order for it to be successful (see https://www.spaces-training.org/), this subtly acknowledges that locals are less capable, or do not have the intellectual capacity to participate in high level science such as that espoused by this journal. The SPACES brief is as follows "It is a collaborative effort between Germany and several southern African countries. Its focus is on the complex interactions between land and sea, and between the atmosphere and biosphere, in southern Africa". "Sustainable research requires early support for young experts in research and education at universities and research institutes, including industry-related ones, in Germany and the participating African countries South Africa, Namibia, Angola, Malawi, Mozambique and Zambia. Good networking, close coordination and long-term cooperation between the scientists of these nations are of equal importance". I do not see this being upheld here.

The skew in funding resources and equipment access further exacerbates this, and the result is a slow whittling of academic resources and ideas from the developing world and further concentration in the developed.

I had written to the editors-in chief to highlight this concern as I feel that in reviewing papers like this, with no clear reasons as to why there are no locals included, I too become part of this growing problem. Their suggestion was to make this comment in my review. On this basis, I cannot pass further judgement on this paper and cannot provide further review or energy on this matter. Simply put, this is just not acceptable anymore and more people need to start standing up to this resurgence of parachute science. Are developing world authors merely to remain as field assistants and people from whom nascent ideas and local access to field sites be sourced from? I sincerely hope not. It is up to us as reviewers, scientists, editors and humans to raise this debate and end this practice.

Links below include the following: https://www.sciencedirect.com/science/article/abs/pii/S0012825220303081 North, M.A., Hastie, W.W. and Hoyer, L., 2020. Out of Africa: The underrepresentation of African authors in high-impact geoscience literature. Earth-Science Reviews, p.103262.

https://onlinelibrary.wiley.com/doi/full/10.1002/esp.5026 Tooth, S. and Viles, H.A., 2021. Equality, diversity, inclusion: ensuring a resilient future for geomorphology. Earth Surface Processes and Landforms.

https://eos.org/articles/why-arent-there-more-journal-papers-by-african-geoscientists?fbclid=IwAR1JdQvhfbc_zFs5jGBQE44IJh4GVn-B_Dm3HwtPdBsfdWieg4_v_1gEzQY

https://www.scientificamerican.com/article/the-problem-of-colonial-science/

And this troubling article spawned by a series of outraged tweets by Brazilian and international scientists https://www.nationalgeographic.com/science/article/one-of-a-kinddinosaur-removed-from-brazil-sparks-legal-investigation

**UbirajaraBelongstoBR search handle on twitter.**

---

## Short Comment (SC1) · 24 Feb 2021

Reply on RC2 by CP editors in chief and handling editor

Anonymous Reviewer 2 was asked to provide a scientific review on the manuscript "Holocene sea level and environmental change at the southern Cape – an 8.5   kyr multi-proxy paleoclimate record from lake Voëlvlei, South Africa" submitted to Climate of the Past.

The Reviewer contacted the Editor and co-Editors in chief to raise the issue of a lack of authors from South Africa in this manuscript, despite the work being part of

a joint German- African "training and knowledge exchange program". This issue, as also expressed by the reviewer, is not a specific authorship issue associated with the manuscript that the editorship of CP can act upon, but instead is a general issue about inclusion and diversity in Science. While it is not the role of the CP-editorship to judge or propose rules related to equity and diversity in research, we do agree that one-way to improve diversity in science is to consider diversity from the start of a project, including providing opportunities for the involvement of early career-researchers and researchers from countries that are currently under-represented in science. This could provide a training and networking opportunity for these researchers, helping them to build a career in science.

It should here be noted however that the authors were not consulted about the issue prior to the submission of the comment by Anonymous Reviewer 2, and we find the accusatory tone of the review very unfortunate. The editor has discussed the question of diversity to the author team and we, editor and co-editors-in chief, are satisfied that in this case the author team provided acceptable replies and is appropriate. We are therefore, in the process of seeking an additional scientific review of the manuscript so that the peer-review process may continue fairly and transparently.

Laurie Menviel, Nerilie Abram, Marit-Solveig Seidenkranz, Denis-Didier Rousseau (CP co-editors-in-chief) and Keely Mills (CP editor)

---

## Author Comment (AC1) · 5 Mar 2021

As a direct response to the specific authorship issue, the author team would like to emphasise that the conducted research and submitted manuscript followed the German Research Foundation (DFG) Code of Conduct Guidelines for Safeguarding Good Research Practice and especially the publication ethics of CP. We absolutely acknowledge the serious topic raised by Anonymous Referee #2. In this context, we would like to emphasise that the author team has ongoing project collaborations and long-term collaborations with African partners, which is reflected in a longstanding and strong networking record of jointly published papers and conference presentations.

---

## Referee Comment (RC3) · Anonymous Referee #3 · 6 Apr 2021

Dear Editor, I have now finished the assessment of the manuscript entitled "Holocene sea level and environmental change at the southern Cape – an 8.5 kyr multi-proxy paleoclimate record from lake Voëlvlei, South Africa", by Strobel et al.

First of all, I would like to remark that I accepted to review this MS only after being assured that the concerns of Reviewer 2 were cleared with the editorial team at Climate of the Past. I thank the journal staff for being vigilant on such practices, and I commend the Copernicus open review system for allowing such discussions to happen.

Abstract. In general, the abstract is clear. I am often in favor for more concise abstracts,

getting to the main conclusions of the study, but I understand this is probably a personal preference. If possible, I would try to shorten the abstract a bit, and make it slightly shorter and more to the point. Minor remarks: Line 17 - "the environmental evoution of..." maybe a part of sentence is missing here. Line 18/19 - Phrasing is odd. It seems that "sea level changes" are archives, which has no meaning. Rephrase and clarify. Line 20 - "it represents an ideal archive". This is referred to the lake, and a lake is not an archive per se, so I would suggest to rephrase.

Introduction. The introduction is generally clear, and presents the state of the art and objectives well. Minor remarks: Line 34 - "future climate projection are even worse". "worse" than what? Line 39 - currents, plural.

Site description. Site description is adequate, giving a good idea of the geographic and geological settings. Minor remarks. Line 86. In Figure 1 you have marked Mossel Bay but not Still Bay. Worth adding?

Material and methods. I am not an expoert on several methods used, but the methodologies closer to my expertise (radiocarbon, OSL, XRD, Grain size) are well described and sound. I would ask the editor to make sure another reviewe can comment on the validity of the other methods employed.

Results. The results are clearly presented and are supported by informative figures. There are some age inversions in the stratigraphy, but the authors present them fairly and give reasonable explanations for their occurrence, also within the discussion.

Discussion. Overall, the discussion is based on the results, and speculation is kept at a minimum. As a side note, my expertise is more focussed on the matters discussed in sections 5.1 and 5.2, but I could follow the discussion in section 5.3 and my opinion is that it is overall sound. One major remark is that I was expecting to see here some discussion of the sites mentioned as "comparison" in Figure 1, but only the GeoB core is described. What about the other records? You should link your text to the sites shown in Figure 1B. Minor remarks: Line 423. In Figure 6, Unit A2 goes to exactly 6ka,

while in the text it is stated that it ends at 6420. Clarify.

Conclusions. The conclusions are short and clear, and adhere to the results and discussions.

Data availability. A data availability statement is missing. I encourage the authors to consult the data availability guidelines of CP and make their data available in an open-access repository. https://www.climate-of-the-past.net/policies/data_policy.html I am also wondering if, given the concerns of Rev.2, it would be good practice to put in the data availability statement the permits / export numbers for the samples that were analysed. Depending on the authority giving the permits, this might also be mandatory.

Figures. Figure 1. Please indicate clearly in the caption that "VOV" is the record you present in this study. Also, are there names associated with the other records? If so, they need to be spelled out in the caption and discussed in the text.

---

## Referee Comment (RC4) · Anonymous Referee #4 · 12 Apr 2021

Dear Editor, I have now finished my review of the manuscript "Holocene sea level and environmental change at the southern Cape, an 8.5kyr multi-proxy palaeoclimate record from lake Voelvlei, South Africa" by Paul Strobel et al.

Agree with reviewer 3, I am pleased that the concerns of reviewer 2 were adequately addressed and the position of the author clarified on the matter, and the matter is now resolved.

My overall thoughts on the criteria set out by CP are given below, comments follow:

This paper represents a substantial contribution to past climates in Southern Africa,

providing novel, new data and a paper that fits well with the scope of the journal. Substantial conclusions are reached based on the data. The abstract could benefit from a sentence or two highlighting the significance of these findings, although

Approaches and applied methods are all valid, with good discussion linking to previous related work, with appropriate references. The scientific methods and assumptions chosen are valid and well outlined, although some contain an over-abundance of detail, and some standard procedures lack references.

Results are presented in a clear, concise way and are well structured. Some suggestions made are that certain elements of the text are restructured for emphasis of these important points. Figures are excellent, although concerns that they will be hard to read if published in their current form are expressed. Grammatical errors and alternative word choices in certain points are given, however the overall story is comprehendible. Some parts (highlighted in the comments) could be clarified or reduced.

Comments:

Title: -

Considering that this is a multi-proxy record reconstructing a range of environmental conditions of which climate is but one factor, it may be more appropriate to describe this reconstruction as a palaeoenvironmental record, rather than palaeoclimate.

Abstract.

The abstract is clear and concise. I suggest putting circa or c. before all dates to indicate the age uncertainty for these events, or to give the full range for the dates.

Line 24: 'Causing' not the right word here

Line 25: Where you say 'moisture', do you mean increased precipitation or wetter conditions? Consider replacing.

Line 28: "Westerly – " is this hyphen supposed to be followed by something, like "-

dominance"?

Line 29: "Climatic spikes". Can you be more specific about what these spikes represent, e.g. do they show climatic deterioration or amelioration?

Line 32: The abstract could use a short summary stating the significance of these findings and the contribution this new data makes towards understanding South African palaeoclimate studies.

Introduction:

Introduction is clear and comprehensive.

Lines 37-45: This section on SA circulation systems occurs a little too early in the manuscript, before the site and study region have been introduced. I suggest moving lines 37 - 45 towards line 50 and edit the text to fit.

Line 46: End the sentence ending with research with something like "in this region".

Line 53: You say "Impact", do you mean impacts, plural? (it would be good to list some, e.g. drainage, fires, grazing).

Line 55: "rock hyrax midden", Use plural "middens".

Line 63: instead of "grain sizes", say grain size analysis, and for pollen, consider changing to palynology.

Line 67: "However" might work better here.

I suggest moving lines 66 "the $\delta2Hn$-alkane signal shows the potential to reconstruct the isotopic signal of precipitation and thus directly refers to the precipitation source..." (and similarly for d13Cn-alkane) so the purpose of these proxies is then followed by their other qualities like preservation potential and past use in the literature for the region, as it is more important for the reader to first know what these proxies are used for.

Line 80: "(in)organic-elemental Be more specific about what method you used here.

Line 82: Your writing tense changes a lot. In this case, you would follow "Specifically we aim to:" with the active tense "Establish". This is the same with the other underlying aims.

Site description:

Line 97: This sentence could use with restructuring, perhaps give specific family or species names for the dominant vegetation types. It is not clear what you mean by "potential", is that because you are unsure or because the vegetation changes from time to time? ps. I'd never heard of Fynbos before and they look incredible

Line 99: "Some pastures persist" This is redundant due to the preceding sentence.

Line 105: "The isotopic composition of precipitation. . ." It would be useful to separate this from the quite general discussions of the environmental conditions for the study region, as this has relevance to your proxy interpretations. Consider including this sentence at the end of that section or in your methods section.

3. Materials and methods:

My area of knowledge is in the age-depth modelling, chronologies and XRF and the methodologies described are sound.

This section needs a sentence stating how you report your ages. In this manuscript, it appears that you state the mid-age range (mean or median) and give the upper and lower 95% CL uncertainty in superscript. Can you clarify this. Also, consider that giving the mid-age range might unintentionally suggest that this is the most likely date for the whole sample. While it is great that you provide this information, it is oftentimes difficult to read.

When introducing each method, include a short sentence or couple of sentences explaining the rationale behind including each of these methods and what they indicate.

It starts too abruptly otherwise.

Line 111: Put the core code-name in brackets. Also on this line, put the figure link (Fig 1:C) in the brackets for the grid reference or state specifically that it refers to the location. It is currently unclear that this does not lead to a figure describing the core.

Line 112: "motor hammer" do you mean a percussion corer?

Line 114: "dark and cool" replace with something like: "under dark and cool ($\sim$4degC) conditions..." and change the plus sign to either a c., $\sim$ or <. Also "split", suggest providing a little detail here. Were they split with power tools and cleaned prior to sub-sampling? Consider saying they were "opened".

Line 115: "sedimentological properties" What method was used, Troel-Smith or an alternative? Perhaps useful to state or cite the standard used. Same as with the sediment colour.

Line 118: "organic macro" do you mean plant-macro? Replace if so.

Line 120: "the AMS..." remove the "the". Its best to spell out abbreviations the first time and put AMS in brackets after and henceforth. Maybe here say "AMS Radiocarbon Facility" or similar.

Line 121: "extraction prior to measurement". Do you mean pre-treatment?

Line 129: "Dose rate analysis" Please provide a reference for this methodology or explain in slightly more detail what this means.

Lines 125-135: Provide a reference for the protocol used or explain the rationale behind each pre-treatment step taken.

Line 135: "Dry-sieving". Give sieve fraction(s).

Line 136: "using the remaining core material" delete this line as it is not needed.

Line 141: "using standard methods"- give methods.

Line 143: "as-measured water content was appropriate" - give a rationale for this assumption.

Line 150: Replace "more than" with >.

Line 164: "results were used". Replace "results" with "dates".

Line 172 "The final age-depth relation was calculated" instead state "profile was modelled..."

Line 192: "Those macrofossils are all snails..." consider replacing with "molluscs", whether terrestrial, freshwater, brackish or marine.

Line 195: "All palaeontological material..." Writing tense changes with this sentence. Is this level of detail necessary? Consider just stating that they will be stored permanently at the Cape Town NHM.

Line 206: What elements are being scanned for? What is your procedure for selecting which elements to use, and what elemental ratios will you use? (provide a rationale and a source for each one). Another method often used is to normalise core-scanning XRF data is to divide the data against the incoherent+ coherent scatter (INC+COH), which are a direct measure of scatter due to these factors. I have not heard of Zr being used before (but I do not work on lakes). Is it not likely that Zr would be affected by these things as much as any other element?

Line 218: State that TOC and TC is reported as percentages.

Line 231-244: Because this is your key proxy, it might be worth bringing this nearer to the top of your methods section.

Line 245: Consider if you need these equations or whether a reference to their source is sufficient

Line 279: Can you explain why these samples gave young ages to justify their exclusion whilst samples deemed stratigraphically correct were retained. Otherwise it comes

across as 'cherry-picking'. What percentage of the dates overlapped with the model? This information is given in Rs console following the model run in Bacon

Results:

The results are clearly described and the figures used to support them do so well (but see comments).

Line 282: It seems from this (and the figure) that a considerable number (6) of the dates do not overlap with the age-depth model

Line 303: What are these ratios used to indicate? Include something about this in the methodology (See: https://www.researchgate.net/publication/281968354_MicroXRF_Core_Scanning_in_Palaeolimnology_Recent_Developm

Line 305: - Remove word "contents". Line 314: "high values in the lower parts" Provide the range of depths, lower parts is too arbitrary

Line 359: "foraminifers" Plural is Foraminiferas or foraminifera, not foraminifers Lines 341-361: It is redundant to keep referring to the figure each time. Just state once that changes in macro and microfossil abundance are illustrated by figure 5.

Discussion:

Discussion based solely on results.

Line 370: "Indicating that sediments were. . ." Consider using "suggesting" as you cannot be certain based on this evidence alone.

Line 389: Ensure that the abbreviation "MAM" is explained at some point within the text.

Line 409: "Summarising" – replace with "in summary. . ."

Line 486: "occasional events" can you be more specific about what is meant by occasional eventa; "amounts" do you mean increased amounts?

Line 540: 'moister' is not a word that is commonly used in this context, typically "more moist" is used but you may want to consider a different word entirely, e.g. wetter

Lines 561- 565: The dating for the youngest flooding event (10 +/- 10 cal BP) would need to have taken place between 1940 and 1960 in order to match with the age-depth model, even considering the dating error. Without using a method specifically designed to accurately date recently deposited surface sediments (e.g. 210Pb or SCPs) this date is highly tenuous and you might want to emphasise this, or state that your age-depth model is likely to be inaccurate towards the top of the profile.

Conclusion: Clear and in line with discussion.

Figures:

Figures are all very informative

Ensure figure 3 & 7 will be readable in a journal article. The current version appears to be low-resolution and it is difficult to make out the detail in the curves, particularly for TIC% which is also very pale for figure 3. Figure 7 is very difficult to read.

---

## Author Comment (AC2) · 8 May 2021

We would like to thank Dr. Keely Mills as the handling editor for the effort with our manuscript, and we thank referees #1, #3 and #4 for their very constructive and valuable feedback which distinctly improved our manuscript. Of course, we considered all reviewer suggestions in the revised version of our manuscript.

**Specific comments to referees**

**Referee #1**

We thank Hayley Cawthra (Referee #1) for her valuable and constructive feedback to our manuscript. We revised our manuscript as suggested.

**Specific comments:**

One broad comment I have, however, is concerning modern-day drought and discussions of past drought events. By the end of the paper, there is a clear history of the relative changes in moisture during the period under consideration, but is it possible to say more explicitly in the text just 'how' dry or how moist the climate may have been, compared to a benchmark of today for example?

The interplay of our proxies enables us to robustly figure out climate variations at Voëlvlei during the past 8.5 ka. However, at the moment we cannot present quantitative statements, but will apply more quantitative approaches in future works.

Lines 19, 23, 78: I am not happy with this term 'intermittent' as it is not clear what you mean. Please define exactly what this refers to in the introduction.

We changed to "At ~6,000 cal BP, the basin of Voëlvlei was filled up with sediment resulting in an intermittent (sporadically desiccated) freshwater lake similar to present."

Line 29: what is a climatic spike?

**We changed to "intense short-term variations".**

Line 31-32: the last sentence of the abstract promises evidence of changing moisture, but the results highlighted in the abstract talk to shifts between easterlies and westerlies. Readers who are not familiar with the area and its climate regime will not be able to infer what that means, so please be more clear with giving a brief summary of moisture fluctuation in the abstract.

**We revised this section as suggested.**

Lines 34-35: this correlation to modern needs for studying past climate needs to be better integrated by the end of the study, as this was the only specific reference I noted to the present conditions. A short section in the discussion, even if only a few lines, to tie back to this 'why we learn about the past' is necessary to loop back to the departure point of the paper.

We include a short section discussing this issue to the revised version of the manuscript.

Line 90: what is the name of the ephemeral river? Is it the Buffels River?

We refer to "Buffels River" in the revised version of the manuscript.

There are a few references missing from the list:

We added the references Chambers et al., 2014 and Rautenbach et al., 2019 to the reference list.

Conolly, 1939, d'Orbigny, 1839, Montagu, 1808, Costa, 1847

These Names and years are not quotations but part of species names giving author and year of first description following the rules of the International Code of Zoological Nomenclature. Thus, we avoided giving these references within the reference list. If the editor regards it being appropriate including these references into the list, however, we will certainly do so.

**Referee #3**

We thank Referee #3 for the valuable and constructive feedback to our manuscript. We revised our manuscript as suggested.

Specific comments:

Line 17 - "the environmental evolution of..." maybe a part of sentence is missing here.

We changed to "South Africa's environmental evolution" to be more precise in the revised version of the manuscript.

*Line* 18/19 - *Phrasing is odd. It seems that "sea level changes" are archives, which has no meaning. Rephrase and clarify.*

We changed to: "Many available sediment archives are peri-coastal lakes and wetlands, however, the paleoenvironmental signals in these archives are often overprinted by sea-level changes during the Holocene."

*Line 20 - "it represents an ideal archive". This is referred to the lake, and a lake is not an archive per se, so I would suggest to rephrase.*

We changed to "it represents an ideal sediment archive".

Line 34 - "future climate projection are even worse". "worse" than what?

We revised the sentence accordingly.

*Line* 86. In Figure 1 you have marked Mossel Bay but not Still Bay. Worth adding?

We added the city of Still Bay to Figure 1B.

One major remark is that I was expecting to see here some discussion of the sites mentioned as "comparison" in Figure 1, but only the GeoB core is described. What about the other records? You should link your text to the sites shown in Figure 1B.

We discuss our data with regard to sea-level and/or paleoenvironmental changes in sections 5.2 and 5.3 and also compare our data in comparison to the studies mentioned in figure 1B.

*Line 423. In Figure 6, Unit A2 goes to exactly 6ka, while in the text it is stated that it ends at 6420. Clarify.*

We revised the figure accordingly.

**A data availability statement is missing.**

We include a data availability statement in the revised version of our manuscript.

Figure 1. Please indicate clearly in the caption that "VOV" is the record you present in this study. Also, are there names associated with the other records? If so, they need to be spelled out in the caption and discussed in the text.

We revised the figure caption and spell-out studies used for comparison now.

**Referee #4**

We thank Referee #4 for the valuable and constructive feedback to our manuscript. We revised our manuscript as suggested.

Specific comments:

The abstract could benefit from a sentence or two highlighting the significance of these findings

We added a statement concerning the significance of our study to the abstract in a revised version of the manuscript.

Considering that this is a multi-proxy record reconstructing a range of environmental conditions of which climate is but one factor, it may be more appropriate to describe this reconstruction as a palaeoenvironmental record, rather than palaeoclimate.

The manuscript title refers to "environmental change" in its first part and we additionally want to note the focus of our study to "paleoclimate changes" in the second part of the title.

The abstract is clear and concise. I suggest putting circa or c. before all dates to indicate the age uncertainty for these events, or to give the full range for the dates.

We revised the abstract as suggested.

Line 24: 'Causing' not the right word here.

We revised the whole sentence.

*Line 25: Where you say 'moisture', do you mean increased precipitation or wetter conditions? Consider replacing.*

We state that moister/wetter conditions occurred at lake Voëlvlei and would therefore prefer to leave this section as written in the current manuscript.

*Line 28: "Westerly –" is this hyphen supposed to be followed by something, like "- dominance"?*

We added "dominance" to be more clear.

*Line 32: The abstract could use a short summary stating the significance of these findings and the contribution this new data makes towards understanding South African palaeoclimate studies.*

We added the following sentence to the abstract: "In contrast to previous investigations, which used indirect proxies for hydrological reconstructions, here we apply a combined biomarkersedimentological-approach that allows a potential identification of precipitation sources, in combination with estimates of moisture availability."

Lines 37-45: This section on SA circulation systems occurs a little too early in the manuscript, before the site and study region have been introduced. I suggest moving lines 37 - 45 towards line 50 and edit the text to fit.

We would like to keep the structure of the text as presented in the manuscript in order to introduce the environmental characteristics forming the year-round rainfall prior the state of research in this area.

Line 46: End the sentence ending with research with something like "in this region".

We changed this as suggested.

Line 53: You say "Impact", do you mean impacts, plural? (it would be good to list some, e.g. drainage, fires, grazing).

We changed to: "[...] impacts, e.g. farming, water abstraction and dredging operations" as discussed in e.g. Haberzettl et al. (2019).

Line 55: "rock hyrax midden", Use plural "middens".

Done as suggested.

Line 63: instead of "grain sizes", say grain size analysis, and for pollen, consider changing to palynology.

Done as suggested.

Line 67: "However" might work better here.

Done as suggested.

I suggest moving lines 66 "the  $\delta^2 H_{n-alkane}$  signal shows the potential to reconstruct the isotopic signal of precipitation and thus directly refers to the precipitation source..." (and similarly for  $\delta^{13}C_{n-alkane}$ ) so the purpose of these proxies is then followed by their other qualities like preservation potential and past use in the literature for the region, as it is more important for the reader to first know what these proxies are used for.

We would like to state the general use of the investigated proxies prior to the site-specific interpretations and therefore prefer to leave the structure of the section of the current manuscript.

*Line 80: "(in)organic-elemental Be more specific about what method you used here.*

We now refer to "assays of inorganic and organic elemental compositions".

Line 82: Your writing tense changes a lot. In this case, you would follow "Specifically we aim to:" with the active tense "Establish". This is the same with the other underlying aims.

**Done as suggested.**

Line 97: This sentence could use with restructuring, perhaps give specific family or species names for the dominant vegetation types. It is not clear what you mean by "potential", is that because you are unsure or because the vegetation changes from time to time?

In this sentence, we refer to the vegetation, which would be present at the site without anthropogenic impacts, i.e. defined as potential natural vegetation. Fynbos is the most diverse vegetation type of the world and consists of >7,000 species (e.g. Manning, 2007). It is hard to identify the most prominent vegetation types of the Fynbos biome potentially growing at the Voëlvlei and we would therefore like to keep our statement as currently written in the manuscript.

Line 99: "Some pastures persist" This is redundant due to the preceding sentence.

**Done as suggested.**

Line 105: "The isotopic composition of precipitation..." It would be useful to separate this from the quite general discussions of the environmental conditions for the study region, as this has relevance to your proxy interpretations. Consider including this sentence at the end of that section or in your methods section.

We are aware that this section is specific to our used proxies. However, it is a general statement of environmental conditions at lake Voëlvlei and we would therefore like to keep the statement at its current position in the manuscript. We additionally refer to the isotopic composition of precipitation in section 5.3.

This section needs a sentence stating how you report your ages. In this manuscript, it appears that you state the mid-age range (mean or median) and give the upper and lower 95% CL uncertainty in superscript. Can you clarify this. Also, consider that giving the mid-age range might unintentionally suggest that this is the most likely date for the whole sample. While it is great that you provide this information, it is oftentimes difficult to read.

We included the following sentence in the revised version of the manuscript: "In the following, ages are reported as median-ages including the upper and lower limit of the 95 % confidence interval."

When introducing each method, include a short sentence or couple of sentences explaining the rationale behind including each of these methods and what they indicate. It starts too abruptly otherwise.

We would like to avoid an introduction of our applied methods in this section due to very site-specific interpretations of several proxies. Instead, we would like to discuss the proxy-interpretation in sections 5.2 and 5.3.

Line 111: Put the core code-name in brackets. Also on this line, put the figure link (Fig 1:C) in the brackets for the grid reference or state specifically that it refers to the location. It is currently unclear that this does not lead to a figure describing the core.

**Done as suggested.**

Line 112: "motor hammer" do you mean a percussion corer?

We modified the sentence as suggested and describe our used system as "percussion hammer coring system".

Line 114: "dark and cool" replace with something like: "under dark and cool (4degC) conditions..." and change the plus sign to either a c., or <. Also "split", suggest providing a little detail here. Were they split with power tools and cleaned prior to sub-sampling? Consider saying they were "opened".

**Done as suggested.**

*Line 115: "sedimentological properties" What method was used, Troel-Smith or an alternative? Perhaps useful to state or cite the standard used. Same as with the sediment colour.*

We followed the standard protocol of the Physical Geography laboratory at Friedrich Schiller University Jena and will add this statement to a revised version of the manuscript.

Line 118: "organic macro" do you mean plant-macro? Replace if so.

We changed as suggested.

Line 120: "the AMS..." remove the "the". Its best to spell out abbreviations the first time and put AMS in brackets after and henceforth. Maybe here say "AMS Radiocarbon Facility" or similar.

Done as suggested.

Line 121: "extraction prior to measurement". Do you mean pre-treatment?

We refer to the extraction of n-alkanes, which is described in section 3.7. Those samples were used for compound-class dating described in this section (3.1.1). We would like to avoid the term "pre-treatment" for the n-alkane samples.

*Line 129: "Dose rate analysis" Please provide a reference for this methodology or explain in slightly more detail what this means.*

This material, as well as representative material from the upper surface immediately adjacent to the sample was dried (24 hours at 105°C) and weighed for water content. It was then homogenised in a ball mill and this powdered material was submitted for determination of its U, Th, and K contents to allow an estimation of the sample dose rate (see below).

Lines 125-135: Provide a reference for the protocol used or explain the rationale behind each pretreatment step taken; Line 135: "Dry-sieving". Give sieve fraction(s); Line 136: "using the remaining core material" delete this line as it is not needed; Line 141: "using standard methods"- give methods.

We revised this section of the manuscript.

*Line 143: "as-measured water content was appropriate" - give a rationale for this assumption.*

In the absence of robust direct evidence for a substantially different water content in the past, we have utilised the modern sample water content. The 3% absolute uncertainty associated with this will account for some fluctuations through time. For reference, a 10% (absolute) change in water content results in an age difference of approximately 700 years.

*Line 150: Replace "more than" with >.*

Done as suggested.

Line 164: "results were used". Replace "results" with "dates".

Done as suggested.

Line 172 "The final age-depth relation was calculated" instead state "profile was modelled..."

Done as suggested.

Line 192: "Those macrofossils are all snails..." consider replacing with "molluscs", whether terrestrial, freshwater, brackish or marine.

The macrofossils described in this sentence are snails. We would therefore like to keep this description.

*Line 195: "All palaeontological material…" Writing tense changes with this sentence. Is this level of detail necessary? Consider just stating that they will be stored permanently at the Cape Town NHM.*

**Done as suggested.**

Line 206: What elements are being scanned for? What is your procedure for selecting which elements to use, and what elemental ratios will you use? (provide a rationale and a source for each one). Another method often used is to normalise core-scanning XRF data is to divide the data against the incoherent+ coherent scatter (INC+COH), which are a direct measure of scatter due to these factors. I have not heard of Zr being used before (but I do not work on lakes). Is it not likely that Zr would be affected by these things as much as any other element?

We added the elements used in this study to the respective section 3.4. Our presented data, i.e. Br/Zr and Al/Zr, were normalised by elemental Zr counts and plotted as log-ratios, primarily to eliminate sediment matrix errors (water content, surface roughness and grain size variations)" as written in the manuscript (c.f., section 3.4). We would have loved to provide COH, INC and/or total counts for normalisation. Unfortunately, the AVAATECH XRF core scanning system (Serial No. 2) used in this study does not provide this.

*Line 218: State that TOC and TC is reported as percentages.*

Done as suggested.

*Line 231-244: Because this is your key proxy, it might be worth bringing this nearer to the top of your methods section.*

We structured our material and methods section from macroscopic to molecular analyses and would like to keep the structure as currently presented in the manuscript.

Line 245: Consider if you need these equations or whether a reference to their source is sufficient

We would like to show these equations for transparency of the calculated proxies.

Line 279: Can you explain why these samples gave young ages to justify their exclusion whilst samples deemed stratigraphically correct were retained. Otherwise it comes across as 'cherry-picking'. What percentage of the dates overlapped with the model? This information is given in Rs console following the model run in Bacon

In section 5.1 we discuss the chronostratigraphy of our record. We added the percentage of overlapping dates to the manuscript, which is 76% when take into account all ages, and 100% excluding ages due to reasons discussed in section 5.1.

*Line 282: It seems from this (and the figure) that a considerable number (6) of the dates do not overlap with the age-depth model*

We discuss the dates not overlapping the 95% confidence interval in our age model in section 5.1.

Line 303: What are these ratios used to indicate? Include something about this in the methodology (See: https://www.researchgate.net/publication/281968354\_MicroXRF\_Core\_Scanning\_in\_Palaeolimnolog y\_Recent\_Developments

The interpretation of the elements is site-specific and therefore we prefer to discuss the elemental ratios in section 5.2 and 5.3 as in the original manuscript.

Line 305: - Remove word "contents".

We would like to differentiate between Al contents/concentrations derived from quantitative elemental analyses (ICP-OES) and semi-quantitative Al/Zr ratios. This is why we refer to Al contents/concentrations and Al/Zr-ratios at this point and would like to keep this in this context.

Line 314: "high values in the lower parts" Provide the range of depths, lower parts is too arbitrary

We added "(< 4.51 m sediment depth)" to the sentence.

Line 359: "foraminifers" Plural is Foraminiferas or foraminifera, not foraminifers Lines

We changed to foraminifera.

341-361: It is redundant to keep referring to the figure each time. Just state once that changes in macro and microfossil abundance are illustrated by figure 5.

Done as suggested.

Line 370: "Indicating that sediments were..." Consider using "suggesting" as you cannot be certain based on this evidence alone.

Done as suggested.

Line 389: Ensure that the abbreviation "MAM" is explained at some point within the text.

We added the explanation to section 4.1.

*Line 409: "Summarising" – replace with "in summary..."*

Done as suggested.

*Line 486: "occasional events" can you be more specific about what is meant by occasional eventa; "amounts" do you mean increased amounts?*

We changed to: "These dry conditions potentially led to a sparse vegetation cover and runoff (induced by occasional events), which likely carried extremely variable grain sizes and amounts of allochthonous input (Al concentration, Al/Zr-ratios)."

*Line 540: 'moister' is not a word that is commonly used in this context, typically "more moist" is used but you may want to consider a different word entirely, e.g. wetter*

We changed to "wetter conditions".

Lines 561- 565: The dating for the youngest flooding event (10 + -10 cal BP) would need to have taken place between 1940 and 1960 in order to match with the age-depth model, even considering the dating error. Without using a method specifically designed to accurately date recently deposited surface sediments (e.g. 210Pb or SCPs) this date is highly tenuous and you might want to emphasise this, or state that your age-depth model is likely to be inaccurate towards the top of the profile.

We changed to: "Considering the chronological uncertainties (dating and modelling errors), the most recent flooding event can likely be associated with the so called 'Laingsburg flood' from January 1981"